# Intercomparison of aerosol measurements performed with multi-wavelength Raman lidars, automatic lidars and ceilometers in the framework of INTERACT-II campaign

Fabio Madonna[1], Marco Rosoldi[1], Simone Lolli[1], Francesco Amato[1], Joshua Vande Hey[2], Ranvir Dhillon[2], Yunhui Zheng[3], Mike Brettle[4], Gelsomina Pappalardo[1].

[1]Istituto di Metodologie per l'Analisi Ambientale, Consiglio Nazionale delle Ricerche (CNR-IMAA)
[2]University of Leicester, Leicester, UK
[3]Sigma Space Corporation, Lanham, MD, US
[4]Campbell Scientific, Shepshed, UK

*Correspondence to*: Fabio Madonna (fabio.madonna@imaa.cnr.it)

**Abstract**. Following the previous efforts of INTERACT (INTERcomparison of Aerosol and Cloud Tracking), the INTERACT-II campaign used multi-wavelength Raman lidar measurements to assess the performance of an automatic compact micro-pulse lidar (MiniMPL) and two ceilometers (CL51 and CS135), respectively, to provide reliable information about optical and geometric atmospheric aerosol properties. The campaign hold at the CNR-IMAA Atmospheric Observatory (760 m asl, 40.60° N, 15.72° E), in the framework of the ACTRIS-2 (Aerosol Clouds Trace gases Research InfraStructure) H2020 project. Co-located simultaneous measurements involving a MiniMPL, two ceilometers, and two EARLINET multi-wavelength Raman lidars were performed from July to December 2016. The intercomparison highlighted that the MiniMPL Range-corrected signals (RCS) show, on average, a fractional difference with respect to those of CNR-IMAA Atmospheric Observatory (CIAO) lidars ranging from 5 to 15% below 2.0 km above sea level (asl), largely due to the use of an inaccurate overlap correction, and smaller than 5 % in the free troposphere. For the CL51, the average fractional difference with respect to the attenuated backscatter of CIAO lidars is <20-30 % below 3 km but larger above. The variability of the CL51 calibration constant is within ±46 %. For the CS135, the performance is similar to the CL51 in the region below 2.0 km asl, while in the region above 3 km asl the differences are about ±40 %. The variability of the CS135 normalization constant is within ±47 %.

Finally, additional tests performed during the campaign using the CHM15k ceilometer operational at CIAO showed the clear need to investigate the CHM15k historical dataset (2010-2016), in order to evaluate potential effects of ceilometer laser fluctuations on calibration stability. The time series of the number of laser pulses shows an average variability of 10 % with respect to the nominal power which conforms to the ceilometer specifications. Nevertheless, laser pulses variability follows seasonal behavior with an increase in the number of laser pulses in summer and a decrease in winter. This contributes to explain the dependency of the ceilometer calibration constant on the environmental temperature hypothesized during INTERACT.

## 1. Introduction

Essential Climate Variables (ECV) accurate monitoring based on the use of low-cost and low-maintenance automatic systems represent one for the scientific community and instrument manufacturers challenges for the for the next decade. The automatic lidars for the vertical profiling of aerosol properties both in the boundary layer and in the free troposphere has reported continuous progress over the last years. Single wavelength elastic backscattering lidars, often with polarimetric capabilities, and ceilometers have the potential to improve our understanding of climate and air quality due to a dense deployment at the global scale (e.g. https://www.dwd.de/EN/research/projects/ceilomap/ceilomap_node.html). Advanced research lidars undoubtedly are

the reference to monitor aerosols, but due to their complexity and high operation and maintenance costs have still a limited geographical coverage. Federated networks set-up by international stakeholders (e.g. GALION – GAW Lidar Observation Network) are slowly evolving towards the harmonization of the different practices adopted within each of the federated networks (e.g. EARLINET, MPLNET, ADNET, LALINET), and, therefore, towards the homogeneity of the respective measurements and products; at present only one example of a coordinated monitoring of a global scale

event (Nabro volcanic eruption) has been provided in literature (Sawamura et al., 2011).

It is useful for the scientific community to understand to which extent automatic lidars and ceilometers (ALCs) are able to provide an estimation of the aerosol geometric and optical properties and fill in the geographical gaps of the existing advanced lidar networks, like EARLINET, the European Aerosol Research Lidar NETwork (Pappalardo et al, 2014). In this direction, at European level, E-PROFILE (http://eumetnet.eu/activities/observations-programme/current-

activities/e-profile/), part of the EUMETNET Composite Observing System (EUCOS), along with EU COST-1303 TOPROF (http://www.toprof.imaa.cnr.it) is spending a large effort to characterize a few of the state-of-the-art ALCs and to establish a good understanding of the instrument output.

Lidars, with respect to the past, evolved into modern automated instruments from strictly research prototypes. Currently, commercial lidar instruments are available on the market and can now efficiently contribute to continuous

monitoring atmospheric aerosol. Automatic lidars have very different features from models equipped with diode-pumped laser or solid-state laser in the UV at 355 nm or in the visible spectrum at 532 nm. Only multi-wavelength lidars emit wavelengths in the near infrared at 1064 nm. Typically, higher the emitted laser pulse energy (spanning from few μJ to mJ), higher will be the required relative maintenance and costs. But higher emitted laser pulse energy translates into higher signal-to-noise ratio that means lower uncertainty affecting the estimation of aerosol properties.

The most important difference between ceilometers and single-wavelength automatic lidars consists in the fact that the former emits a single wavelength in the near infrared between 900 and 1100 nm to avoid strong Rayleigh scattering with a pulse repetition rate of the order of a few kilohertz and laser pulse energy of few μJ, to allow eye-safe, continuous and unattended operations. UV and visible automatic lidars can typically cover the whole tropospheric range, while ceilometer, depending on the model, can cover the boundary layer only or detect aerosol features also in

the free troposphere.

Limitations in aerosol property retrievals by different ceilometers have been already investigated (e.g. Wiegner et al. 2014, Madonna et al., 2015, Kotthaus et al., 2016). Ceilometers are limited to retrieve the attenuated backscatter and the aerosol backscattering coefficient with a limited accuracy. For the latter, the retrieval is affected by the calibration of the aerosol backscattering profiles and relies on the use of ancillary instruments, such as a co-located Raman multi-

wavelength lidar or a sun photometer, or, depending on the ceilometer model, can be performed using the molecular backscattering profile in an aerosol-free region (only by adopting long integration time, larger than 1-2 hours, depending on the atmospheric conditions; Wiegner et al., 2014). Alternatively, ceilometers can be calibrated following the procedure described in O'Connor et al. (2004), where the backscatter signal is rescaled until the observed lidar ratio value matches the theoretical value, when suitable conditions of stratocumulus are available. In addition, ceilometers

use diode laser sources working in an infrared region where the water vapor absorption is strong. At those wavelength regions, a correction of the profiles using a radiative transfer model is mandatory for retrieving optical properties (Wiegner and Gasteiger, 2015).

Given the role that commercial lidars and ceilometers might cover due to their low-cost and low-maintenance baseline component of the aerosol non-satellite observing system at the global scale, several intercomparison experiments must

be designed to assess the performances of commercial systems compared to advanced multi-wavelength lidars and to ensure comparability between different instruments, measurements and retrieval techniques. These experiments can provide recommendations which can strongly support the design of current and future networks for the aerosol observation and the monitoring of pollution.

Behind this motivation, the INTERACT campaign was arranged and took place at CIAO (CNR-IMAA Atmospheric

Observatory) in Tito Scalo, Potenza, Italy (760 m asl, 40.60°N, 15.72°E) from July 2014 to January 2015 (Madonna et al., 2015). It demonstrated good performance of the ceilometers using diode-pumped Nd:YAG lasers, like the CHM15k type, but also pointed out difficulties using the molecular calibration to retrieve aerosol properties. The variability of the ceilometer calibration constant, calculated using an advanced multi-wavelength Raman lidar as the reference, requires a frequent monitoring of the calibration at minimum on a seasonal basis. Thermal effects along with a non-linear system

response to different aerosol loadings have been considered the potential reason for the Nd:YAG ceilometers' instability.

With the same INTERACT general campaign objectives, i. e. to provide a continuous investigation of the automatic lidar and ceilometer performances, the INTERACT-II campaign has been performed at CIAO from July 2016 to January 2017 in the framework of the transnational access activities of the H2020 research infrastructure project
ACTRIS-2 (Aerosol Clouds Trace gases Research InfraStructure, http://www.actris.eu). During this period, different pure or mixed aerosol types were observed at CIAO both in the the boundary layer and in the free troposphere, such as mineral dust, biomass burning, continental, rural and pollution. Aligned to those of INTERACT, the main scientific objectives of INTERACT-II have been to:

➢ Evaluate the performance of commercial automatic lidars and ceilometers to retrieve the geometric and optical aerosol/cloud properties (with respect to the instrument sensitivity to different loads and types of aerosols and clouds);

➢ Assess the instrument Signal to Noise Ratio (SNR) and dynamic range (depending on the aerosol extinction coefficient, water vapor content, solar irradiance, etc.);

➢ Study the instrument stability over time (e.g. laser, detector, efficiency, thermal drifts, etc.);

➢ Assess the ceilometers' calibration stability and accuracy (using ACTRIS/EARLINET Raman lidars as a reference).

The campaign included an automatic lidar (MiniMPL, provided by Sigma Space Corporation), and four ceilometers (Campbell CS135, VAISALA CT25K and CL51, and Jenoptik CHM15k).

INTERACT-II adopted INTERACT (Madonna et al., 2015) campaign philosophy and methodological approach with
the added value to intercompare at once the newest generation of 905-910 nm ceilometers, the MiniMPL lidar, recently delivered on the market, and the advanced multi-wavelength Raman lidars operated at CIAO, including the EARLINET reference mobile system, MUSA (Multi-wavelength System for Aerosol). The capability of the MiniMPL and ceilometers to detect aerosol layers and provide quantitative information about the atmospheric aerosol geometric and optical properties was investigated. Advanced Raman lidar measurements are provided by the two permanently
deployed lidars operative at CIAO: MUSA, which is one of the mobile reference systems used in the frame of the EARLINET Quality Assurance Program, and PEARL (Potenza EArlinet Raman Lidar). Range corrected signals (RCS) of CIAO Raman lidars (hereinafter CIAO LIDARs) have been compared with those provided by the MiniMPL lidar, while the CIAO LIDAR attenuated backscatter coefficient profiles (β') have been compared with the corresponding β' profiles provided by ceilometers.

CHM15k and CT25K performances shown during INTERACT-II are not discussed in this paper because both the ceilometers have been already characterized during INTERACT. In addition, the CHM15k underwent through a laser realignment from July to October 2016 and the system has been mainly used during the last part of the campaign to perform a few stability tests of the laser which are described later on in the paper.

In the next section, we describe the instruments deployed during INTERACT-II. In section 3, the algorithms used for
the data processing are presented. In Section 4, we show and discuss the intercomparison results between CIAO LIDARs and MiniMPL, while ceilometers' performances are described in Section 5. The stability of the ceilometers with respect to the changes in the environmental temperature is analyzed in Section 6. Summary and conclusions are finally provided.

## 2. Instruments

Located in the middle of the Mediterranean region, with proximity to the sea at less than 150 km in most directions and located in a strategic location with respect to the occurrence of African dust outbreaks and Eastern European forest fires, CIAO represents an ideal location for different aerosol species observations under different meteorological

conditions. Beyond the multi-wavelength Raman lidars and the ceilometers mentioned in the introduction, CIAO utilizes a suite of instruments that provides continuous observation of the atmosphere, including a microwave radiometer, a Ka-band cloud radar, a sun-star-lunar photometer. Moreover, radiosoundings are launched weekly (Madonna et al., 2011).

Ceilometers were installed on the roof of the observatory building (about 10m above the ground), while the MiniMPL, which is heavier and larger than a ceilometer, has been deployed close to MUSA and PEARL at the surface. Table 1 reports the specifications of the MiniMPL, MUSA and PEARL at 532 nm, while Table 2 provides specifications for the infrared receivers of ceilometers, MUSA and PEARL.

MUSA is a mobile multi-wavelength lidar system based on a Nd:YAG laser source at 1064nm, that it is doubled and tripled to add additional wavelengths at 532 and 355nm. The receiver unit consists of a Cassegrain telescope with a primary mirror of 300 mm diameter. The three laser beams are simultaneously and coaxially transmitted into the atmosphere beside the receiver in biaxial configuration. The receiving system has 3 channels to detect the elastically backscattered radiation from the atmosphere and 2 additional channels for Raman inelastically backscattered radiation by atmospheric $N_2$ molecules at 607 and 387 nm. The elastic channel at 532 nm is split into parallel and perpendicular polarization components by means of a polarizing beam splitter cube. The backscattered radiation at all the wavelengths is acquired by photomultiplier tubes both in analog and photon counting mode. The calibration of depolarization channels is automatically made using the ±45 method (Freudhentaler et al., 2009). The typical vertical resolution of the raw profiles is 3.75 m at 1 min temporal resolution. The MUSA system is compact and transportable and it is one of the reference systems employed for the EARLINET quality assurance program. MUSA is routinely tested with respect to several systematic quality-assurance tests developed in order to harmonize the lidar measurements, setting up high quality standards and improving the lidar data evaluation (Pappalardo et al., 2014). MUSA signals are also routinely evaluated using the Rayleigh fit test, and signal-to-noise analysis (Baars et al. 2016). Additionally, the telecover test (Freudenthaler, 2008) is performed regularly and especially after transportation of the system. The system is aligned using a CCD camera to reduce the effect of misalignment between the telescope and laser axis, being MUSA a bistatic lidar. Finally, the multi-wavelength detection capability enables the so called "3+2" lidar data analysis which, taking advantage of the simultaneous retrieval of aerosol extensive (extinction coefficients at 355 nm and 532 nm; backscattering coefficients at 355 nm, 532 nm and 1064 nm) and intensive optical properties (lidar ratios at 355 nm and 532 nm and color ratios) at different wavelengths, permits to check the physical consistency of the retrieved aerosol properties.

The multi-wavelength lidar system for tropospheric aerosol characterization, PEARL (Potenza EArlinet Raman Lidar), has been designed to provide simultaneous multi-wavelength aerosol measurements for the retrieval of optical and microphysical properties of atmospheric particles as well as water vapor mixing ratio profiles. The system, operated according to regular EARLINET measurement schedule until 2014, is presently used only for testing, during special events, and as back-up for the MUSA system when MUSA was moved abroad for the calibration of the EARLINET stations (Wandinger et al., 2016). PEARL is based on a 50 Hz Nd:YAG laser source emitting at 1064, doubled and tripled to 532 and 355 nm, respectively. An optical system based on mirrors, dichroic mirrors and 2X beam expander separates the three wavelengths allowing optimization of the energy and divergence for each wavelength. The beams are mixed again for collinearity of the three wavelengths and transmitted simultaneously and coaxially with respect to the lidar receiver. The backscattered radiation from the atmosphere is collected by an F/10 Cassegrain telescope (0.5m diameter, 5m focal length) and forwarded to the receiving system, where three channels detect the radiation elastically backscattered from the atmosphere at the three laser wavelengths and three channels are used for the Raman radiation backscattered from the atmospheric $N_2$ molecules at 387 nm and 607 nm and from $H_2O$ molecules at 407 nm. Two additional channels detect the polarized components of the 532 nm backscattered light. Each of these channels is further split into two channels differently attenuated for the simultaneous detection of the radiation backscattered from the low and high altitude ranges, in order to extend and optimize the signal dynamic range. For the elastic backscattered radiation at 1064 nm the detection is performed by using an Avalanche Photo Diode (APD) detector and the acquisition is performed in analog mode. For all the other acquisition channels, the detection is performed by means of photomultipliers and the acquisition is in photon-counting mode. The vertical resolution of the raw profiles is 7.5 m for 1064 nm and 15 m for the other wavelengths, and the raw temporal resolution is 1 min. PEARL measurements were extensively intercompared with MUSA to have a redundant aerosol profiling capability at CIAO.

The MiniMPL transceiver weighs 13 kg and measures 380 × 305 × 480 mm (width, depth, and height). The system consists of a laptop and the lidar transceiver, connected by a USB cable, and the average power consumption is about 100 W during normal operations. The whole system fits in a transportable storm case with a telescopic handle and wheels and can be checked in as regular luggage during a domestic or international flight. The MiniMPL's Nd:YAG laser emits polarized 532 nm light at a 2.5 KHz repetition rate and 3.5-4 µJ nominal pulse energy. The laser beam is expanded to the size of the telescope aperture (80 mm) to satisfy the eye safe requirements in ANSI Z136.1.2000 and IEC 60825 standards. The system also has built-in depolarization measurement (Flynn et al., 2007) with a contrast ratio greater than 100:1. The receiver uses a pair of narrowband filters with bandwidth less than 200 pm to reject the majority of solar background noise. The filtered light is then collected by a 100 µm multimode fiber and fed into a Silicon Avalanche Photodetector operating in photon-counting mode (Geiger mode). Photon-counting detection enables the MiniMPL design to be lightweight and compact with high signal-to-noise ratio (SNR) throughout the troposphere. MiniMPL sets the laser beam divergence at about 40 µrad and receiver Field-Of-View (FOV) at 240 µrad. This design balances the solar noise with optical system stability and avoids multiple scattering which can distort measurements of depolarization ratio and extinction coefficient in the cloud.

The Vaisala Ceilometer CL51, the second generation of Vaisala single lens ceilometers, is designed to measure high-range cirrus cloud base heights while maintaining the capability to measure low and middle range clouds and, in high turbidity conditions, to diagnose vertical visibility. Its application to detection of tropospheric aerosol layers is under investigation in several papers in literature (e.g. Wiegner et al., 2014). The CL51 employs a pulsed diode laser source emitting at 910±10 nm (at 25 ∘C with a drift of 0.27 nm $K^{-1}$) with a repetition rate of 6.5 kHz. The refractor telescope, that employs an enhanced single lens technology, theoretically allows reliable measurements virtually at the surface, although the overlap correction estimated by the manufacturer is not able to effectively correct the ceilometer profile over the entire incomplete overlap region. The backscattered radiation is filtered using an optical bandpass filter which, according to Vaisala, is in the order of 3.4 nm and then detected using an APD in analog mode. The instrument used in INTERACT-II was updated with the latest firmware version (v1.034).

The Campbell scientific CS135 ceilometer employs a pulsed diode laser source emitting at 912±5nm with a repetition rate of 10 kHz. The ceilometer receiver is based on a single lens telescope. Half of the lens is used for the transmitter and the other for the receiver with a total optical isolation between them. The optical layout is conceived to enable lower altitude measurement and to integrate larger optics into a compact package. Like the CL51, the backscattered radiation is filtered using an optical bandpass filter (36 nm) and detected using an APD in analog mode. The latest version of the instrument firmware was provided by the manufacturer itself. During INTERACT-II, CS135 data collection (performed using a terminal emulator) was affected by a technical problem with the CIAO logging system, which caused the loss of a large amount of data especially in the free troposphere, thus limiting the number of available cases for the comparison (only 9 measurement sessions).

At this stage, it is worth providing a few clarifications about the hybrid nature of this intercomparison campaign which involved both automatic elastic (polarized) lidars and regular ceilometers. As remarked upon in Madonna et al. (2015), ceilometers are optical instruments based on the lidar principle, but eye-safe and generally lower in cost and performance compared to advanced research or automatic elastic lidars. Their primary application is the cloud base height determination and vertical visibility for transport-related meteorology applications. These instruments typically have considerably lower SNRs than lidars because they employ diode lasers and wider optical bandpass filters to detect over the broader spectrum of these sources. Diode lasers sources are often permitted only if they observe eye-safety limits which permits ceilometers to be operated unattended. In a few more powerful ceilometers, like the CHM15k and CHM15kx, as well as the MPLs (including MiniMPL), the use of diode-pumped lasers allows much larger SNRs and, therefore, enhanced performances (e.g. Madonna et al., 2014). Moreover, ceilometers, while providing factory calibrated attenuated backscatter profiles, do not often provide the raw backscattered signals and their processing software includes several automatic adjustments of the instrument parameters (e.g. gain, voltages, background suppression, etc.) performed according the observed scenario (e.g. daytime, night time, clear sky or cloudy) but out of the control of users. This makes it difficult to use them for research purposes beyond the applications for which they were designed.

During INTERACT-II, a hybrid ensemble of these instruments, automatic lidars and ceilometers have been deployed. Nevertheless, the main scope of the campaign remains the assessment of the performances of each different category of

instruments separately, and, within the same category, to assess the limitation in the use of each system involved.
Therefore, the results presented in section 4 and 5 are intended to show under which limitations each of the investigated systems is able to provide quantitative information on the aerosol properties in both the boundary layer and in the free troposphere. The reader should use these results according to his or her own specific needs and with careful consideration of the application.

**3.  Intercomparison methodology and data processing.**

Following the same approach used during INTERACT, CIAO LIDAR signals have been processed using the EARLINET Single Calculus Chain (SCC) (D'Amico et al., 2016; Mattis et al., 2016). The SCC outputs are the pre-processed range corrected signals (RCS) and the profiles of aerosol extinction coefficient at 355 and 532 nm and backscattering coefficient at 355, 532 and 1064 nm, using both Raman and elastic signals. RCS is defined as the
product of the pre-processed signal (background subtracted) multiplied by the square of the altitude range: RCS= $P(z)$ $z^2$, where $P(z)$ is the lidar pre-processed signal and z is the altitude range for a zenith pointing lidar.

In contrast to the ceilometers, the MiniMPL also provides the raw signals, acquired in photon counting mode only, enabling the direct comparison with the CIAO LIDAR signals. RCS is a quantity proportional to the attenuated backscattering β', which is used for the investigation of ceilometer performance and is defined as:

$$\beta' = \frac{P(z)z^2}{C_L} = \beta(z)T^2(z) \quad [\text{Eq.1}]$$

where $C_L$ is the lidar constant (depending only on the lidar experimental setup), $\beta(z)$ is the total (aerosol plus molecular) backscattering coefficient, and $T^2(z)$ is the two-way transmissity of the atmosphere. The use of RCS allows a comparison between the two systems over a vertical range larger than the range where β' is available. This is because the β' calculation depends on the range covered by the retrieval of the CIAO LIDAR extinction coefficient using the
Raman method, applied in this work. The lower SNR typical of the Raman lidar channels does not allow to provide a vertical profile of the aerosol extinction coefficient over the entire range typically covered by an elastic lidar signal. The use of RCS brings the comparison to the signal level, avoiding calculation of higher level products, whose retrieval can increase the number of assumptions and uncertainties (e.g. Lolli et al., 2017).

To perform the comparison between CIAO LIDARs and MiniMPL, 532 nm MiniMPL RCS is normalized to the
corresponding CIAO LIDAR RCS, on a profile-per-profile basis for every profile, over a vertical range of 1.2 km starting from a variable reference altitude between 6 and 8 km asl, where the aerosol content is identified as negligible qualitatively using quicklooks of the lidar time series. All the time series considered in this comparison refer to night time clear sky measurements. The profiles from all the instruments are compared over a vertical resolution of 60 meters and a temporal integration time ranging from 1 to 2 hours, selected automatically by the SCC depending on the
observed atmospheric scenario. No vertical smoothing is applied to the data, but systems outputting data at a higher resolution are interpolated to the CIAO LIDAR resolution. All of the profiles are cut in the lower part of the atmosphere, below 1300 m asl, in order to consider CIAO LIDAR reference lidar signals only in the region with the full overlap between the telescope and laser beam. The number of the simultaneous CIAO LIDARs and MiniMLPL measurements time series has been limited by a few periods of unavailability of the MiniMPL due to an issue in the
regulation of the instrument housing temperature.

For the ceilometers, the comparison was carried out using the 1064 nm β' profiles obtained through their normalization over the corresponding CIAO LIDAR β' profile below 3 km asl over a vertical range of 600 m, where the full overlap of all instruments was ensured. Given that ceilometer measurements are performed at 910-912 nm, β' profiles have been rescaled using the 532/1064 backscatter-related Ångström coefficient measured by CIAO LIDARs in order to obtain the
equivalent profile at 1064 nm for comparison with CIAO LIDARs. For those altitudes where the backscatter-related Ångström coefficient was not available (typically in the free troposphere (FT), above 5 km asl) a climatological value of 1.05 was used. The uncertainty contribution for the spectral dependence of β' and, therefore, of the aerosol backscattering coefficient and of molecular and aerosol extinction coefficients has been estimated within a few percents. More details on calibration are discussed in section 5.

A ceilometer β' profile can only be retrieved if water vapor absorption is taken into account (Wiegner et al., 2015). The influence of water vapor absorption at operating wavelengths of ceilometers is due to the presence of a strong absorption band between 900 and 930 nm, while at 1064 nm there is no absorption. Therefore, the retrieval of β' profiles must consider the attenuation of the backscattered radiation by water vapor. In this study, the method used for correcting the attenuation by water vapor is based on the Fu-Liou-Gu (FLG) radiative transfer model (Gu et al., 2011),
in the modified version discussed in Lolli et al. (2017b).

    FLG is a combination of the delta four-stream approximation for solar flux calculations (Liou, 1986) and a delta-two-four-stream approximation for IR flux calculations. The solar (0–4 μm) and IR (4–50 μm) spectra are divided into 6 and 12 bands, respectively, according to the location of prominent atmospheric absorption bands. FLG makes use of the adding procedure to compute the spectral albedo in which the line-by-line equivalent radiative transfer model (Liou et
al. 1998) uses the correlated k-distribution method for the sorting of absorption lines in the solar spectrum. In the solar spectrum, non-gray absorption due to water vapor, $O_3$, $CO_2$, $O_2$, and other minor gases, such as CO, $CH_4$, and $N_2O$, is taken into account. Non-gray absorption due to water vapor, $O_3$, $CO_2$, $CH_4$, $N_2O$ and CFCs is considered in the IR spectrum. Potenza GRUAN (GCOS Research Upper-Air Network) processed (collocated) radiosoundings were used as input for the FLG radiative transfer model (Lolli et al., 2017a) in about 40% of the cases, while for the remaining cases,
when local radiosoundings were not available, data from closest RAOB (The Universal RAwinsonde OBservation program) site located in Brindisi Casale (40.63N, 17.94E, 15 m), about 150 km east of Potenza, were used. RAOB profiles were cut at the CIAO altitude level (760 m). According to the correction method suggested in literature for 905-910 nm ceilometers (Wiegner et al., 2015), an optimal correction would require the knowledge of both the laser wavelength and the bandwidth for each emitted pulse. These data are not currently stored and provided by the
ceilometer hardware. Therefore, to estimate the water vapor correction a laser Gaussian profile centered at the nominal laser wavelength with FWHM (Full Width Half-Maximum) of 3.5 nm has been assumed. Moreover, FLG has a spectral resolution of 50 $cm^{-1}$, while in literature a resolution lower than 0.2 $cm^{-1}$ is recommended to avoid an "unpredictable" behaviour of the model calculation. The water vapor absorption has been calculated through the average absorption within the spectral range described above. In addition, the comparison between the ceilometers and the lidars, discussed
in section 5, shows that the uncertainty due to the water vapor correction cannot represent the main contribution to the total uncertainty budget of 905-910 nm ceilometer measurements.

    For the comparison between CIAO LIDARs and MiniMPL, it is important to remark that MUSA detects with two channels the co- and cross polarized components of the elastically backscattered radiation at 532 nm, in order to measure the particle depolarization at that wavelength. MiniMPL also detects the co- and cross polarized components of
the elastically backscattered radiation at 532 nm and provides continuous measurements of particle backscattering coefficient and depolarization ratio profiles. Because of different polarization setups, MUSA measures the particle linear depolarization ratio (Freudenthaler et al., 2009) while mini-MPL measures the particle circular depolarization (Flynn et al., 2007). For both MUSA and MiniMPL, total signals must be calculated for through the combination of the respective co- and cross-polarized channels. 532nm MiniMPL RCS has been calculated according to the equations
provided in Campbell et al. (2002). PEARL, instead, is equipped not only with the co- and cross-polarized channels at 532 nm, but also with channels detecting the 532 nm total backscattered radiation.

    To provide a first example related to the dataset discussed in this paper, a comparison of the 532 nm PEARL RCS and MiniMPL RCS at their own time and vertical raw resolutions is shown in Figure 1 for the measurements collected on 13 October 2016 from 18:00 to 19:00 UT. Figure 2 shows the comparison of the 1064 nm PEARL RCS with the 910-912
nm CL51/CS135 attenuated backscatter for the same day. To ensure correct interpretation of Figures 1 and 2, it is important to reiterate that raw time and vertical resolutions are 1 minute and 15 m for PEARL, 5 minutes and 30 m for MiniMPL, 30 seconds and 10 m for CL51, 30 seconds and 5 m for CS135.

    Finally, it is also important to note that the CIAO operator routinely checked each instrument during INTERACT-II to ensure that each one was performing according to the manufacture specifications. The routine maintenance included:

a.   A daily inspection of each instrument and its operation;

    b.   A weekly check on each instrument's acquisition parameters (laser transmitter, receiver, heater, blower, windows, tilt angle, etc.);

c.  Approximately bi-weekly cleaning of the windows, with frequency depending on atmospheric conditions (e.g.
after precipitation or dust/smoke transport events), using the flooding method. Additionally, specific treatments
        to remove the stronger dust spots were performed in response to warning messages provided by each
        instrument (e.g. window contamination messages);

d.  Dark current measurements were made twice during the campaign for ceilometers, using a termination hood
        provided by the manufacturer while operating in analog detection mode. Dark current profiles were subtracted
from each of the raw backscatter profiles before normalization using the lidar; for MUSA and PEARL, dark
        currents were routinely estimated before each measurements session.

## 4.  MiniMPL vs MUSA: Comparison of Range-corrected signals

Simultaneous observations of aerosol collected with the multi-wavelength Raman lidars operative at CIAO, MUSA and
PEARL, and of the automatic Sigma Space mini-MPL, collected during the measurement campaign, have been
compared.

An example of comparison between RCS provided by MUSA and mini-MPL is shown in the left panel of Figure 3,
related to the observations collected on 29 August 2016 from 19:16 to 20:47 UT. The quicklooks of the RCS time series
(not reported) show a sharp aerosol layer between about 1.5 km and 2.5 km asl along with a lower aerosol loading
below the layer to the ground, while the atmosphere is dominated by the molecular scattering above. In the right panel
of Figure 3, the air mass back trajectory analysis performed using the NOAA HYSPLIT (Hybrid Single Particle
Lagrangian Integrated Trajectory) model (Stein et al., 2015) initialized at three levels from the ground to the top height
of the highest layer observed by both MUSA and MiniMPL lidars. Trajectories are obtained using the vertical velocity
model of HYSPLIT running the back-trajectories for a length of 200 hours at three vertical levels.

The difference between the two profiles shows a good agreement throughout the troposphere with discrepancies < 5%
between 2.0 km and 4.0 km asl, within the RCS random uncertainty (D'Amico et al., 2016). MiniMPL underestimates
MUSA (up to 10% RCS) at altitudes lower than 2.0 km asl, in the incomplete overlap region. MiniMPL data processing
provides a correction function which is not able to properly adjust all of the collected signals in the incomplete overlap
region. The beam pointing instability of the laser in this vertical range is likely the reason preventing the adjustment
using a precomputed static correction function.

A second example (left panel of Figure 4) shows RCS valued provided by MUSA and mini-MPL collected on 04 July
2016 from 19:56 to 21:45 UT. Multiple aerosol layers up to 4.0 km asl are observed. In the right panel of Figure 4, the
corresponding air mass back trajectory analysis shows the quasi-zonal transport of the observed aerosol from North-
East Canada over the Atlantic Ocean to Europe. Also in this case, the comparison shows a good agreement throughout
the troposphere with discrepancies <5%, which are identified both in the incomplete overlap region and above this
region and up to 4.0 km of altitude, where most of the aerosol loading is located. This might be related to the
uncertainty affecting the estimation of corrections other than overlap applied to the MiniMPL data processing, e.g.
after-pulse correction. The manufacturer shall investigate this hypothesis. Nevertheless, the discrepancies are within the
RCS random uncertainty and do not compromise the good agreement between the two systems.

In Figure 5, the black line shows the profile of the average fractional difference between CIAO LIDAR and MiniMPL
values of RCS calculated for 12 cases of simultaneous and collocated measurements collected in the period from July to
December 2016. The vertical bars are the standard deviations of fractional differences. Fractional difference is defined
as the relative difference between CIAO LIDAR RCS and MiniMPL RCS values with respect to normalized by CIAO
LIDAR RCS. The profile shows that MiniMPL underestimates CIAO LIDARs MUSA in the region below 2.0 km with
an increasing average fractional difference towards ground level; the maximum value of this deviation is less than 15%.
The blue line reported in Figure 5 represents the same as the black line but adjusted by applying an additional overlap
correction factor to the MiniMPL, estimated using the ratio between MUSA and MiniMPL RCS profiles during the
cleanest simultaneous measurement session available during INTERACT-II. The additional correction applied from the
ground to 3.3 km asl, identified as the overlap height for the MiniMPL, reduces the average fractional difference in the
range from 1.5 km to 3.3 km, with values less than 3% from 1.8 km and the standard deviation of the difference keeps
to within 10%. Below 1.5 km, the correction is not able to properly adjust the profile due to the presence of the aerosol

residual layer in the measurements used to estimate the correction factor. The example correction for the overlap effects provided in Figure 5 cannot be considered exhaustive, but demonstrates that some work is required to improve the MiniMPL data processing in the incomplete overlap region. In the remainder of this section, the MiniMPL original data processing will be considered.

To evaluate the MiniMPL stability during the campaign, the values of the normalization constant referred the two periods when MUSA and PEARL, respectively, have been used as the reference lidar, have been separately averaged to calculate a relative variability for the same constant. The normalization was typically performed between 6 km and 8 km asl. Then the average of the two calculated relative variabilities for these two different periods has been calculated showing that the stability of the MiniMPL calibration ("lidar normalization") during the campaign was within ±29 %. This value embeds the PEARL-MUSA system variability which is evaluated from the molecular calibration constant and it is for both the systems within 20%. However, given both the number of simultaneous observations available and the use of two lidar systems as the reference lidars in two different time periods, the estimation of the calibration stability must be handled with caution. In general, the MiniMPL showed a good stability in its operation in the considered time period and with respect to seasonal changes in the environmental temperature and in the aerosol loading.

In Figure 6, the comparison between CIAO LIDARs and MiniMPL probability density functions (pdfs) of the RCS values confirms the overall good agreement between CIAO LIDARs and mini-MPL, with some tendency of mini-MPL to overestimate CIAO LIDARs for RCS values lower than $1.5 \times 10^{10}$ (a.u.): this difference is more evident in the left panel of Figure 6, where pdfs are calculated for the vertical range below 4.0 km asl.

Finally, in Figure 7, the relationships between the 532 nm aerosol (particle) extinction coefficient ($\alpha_{par}$) from MUSA and PEARL lidars and the corresponding RCS at 532 nm measured by MUSA and PEARL lidars and by MiniMPL is shown to highlight differences in lidar sensitivity to different aerosol extinction coefficients. $\alpha_{par}$ is calculated over the same temporal window as RCS, but with a lower effective vertical resolution (typically within 480-600 m) in order to reduce the uncertainty and the related oscillation affecting the extinction profile calculated using the Raman lidar signal. The output profile vertical resolution is 60 m to match the RCS vertical resolution. The comparison in Figure 7 shows a good agreement between MiniMPL and CIAO LIDARs. Small differences can be identified and are more evident for values of $\alpha_{par}$ larger than about $5.0 \times 10^{-5}$ m$^{-1}$, where MiniMPL RCS values are more scattered compared to CIAO LIDARs. The RCS differences may be the results of systematic effects due to inaccurate adjustments applied to the signal processing, including the incomplete overlap correction, which for MiniMPL looks quite relevant in the region between 1.0 km and 3.3 km asl.

## 5. Ceilometer: Comparison of attenuated backscattering

This section focuses on the comparison of the attenuated backscatter ($\beta$') simultaneously measured by MUSA and PEARL multi-wavelength Raman lidars and estimated for CL51 and CS135 ceilometers. The left panel of Figure 8 shows the attenuated backscatter retrieved from PEARL, CL51 and CS135 on 13 October 2016 in the time interval from 17:47 to 19:08 UT. The HYSPLIT air mass back trajectory analysis (not shown) reveals that the observed advected aerosol layers may come from Libya and Morocco, two regions where large sources of dust are present for the different altitude levels where aerosol layers are observed with MUSA. The agreement between the three instruments is extremely good below 2.5 km asl. Between 2.5 and 3.7 km asl the differences are larger for both the CL51 and the CS135 (larger difference shown by CS135). The difference between the CL51 and CS135 in the region between 2.5 km and 3.5 km asl may be also partly affected by the dependency of the water vapor correction on the emitted laser spectrum. The CS135 signal strongly decreases above 3.5 km close to the top region of the second observed aerosol layer. The CL51 signal is higher but the noise suggests that it is not reliable to detect the residual aerosol backscattered radiation at that altitude range as well the molecular return. All the CL51 profiles shown in Figure 8 are cut below 5.0 km asl, to better visualize the comparison otherwise affected by the large noise oscillation of the signals.

The right panel of Figure 8 shows attenuated backscatter measured by the same instruments on 01 December 2016 from 17:53 to 19:19 UT. The air mass backtrajectory analysis for this time period showed that the observed air mass originated in Canada and reached CIAO via North-West Europe. This comparison reveals the effect of ceilometer variability in the region of incomplete overlap: the correction applied by the manufacturer is often able to adjust the profile minimizing the difference with respect to the reference CIAO LIDARs, but in many other cases, as for 01

December, differences are considerable. It is worth reiterating that, as for the MiniMPL comparison, all the profiles are cut off below 1.3 km asl because CIAO LIDARs are considered as a reference only in the full overlap region.

Regarding the CL51 β' profiles, the choice of normalization range has proven to be more critical than expected. Initially, all the CL51 profiles were normalized over a 0.6 km vertical range below 8 km asl, in order to find a trade-off between an acceptable CL51 SNR and the need to normalize in a stable aerosol free region of the atmosphere. Nevertheless, the CL51 SNR is too low in the FT and the decrease in its sensitivity to the molecular return makes the normalization to the lidar in the FT (and consequently the ceilometer molecular calibration) challenging. The left panel
of Figure 9, shows the comparison between β' retrieved from MUSA and CL51 on 4 July 2016 from 19:56 to 21:45 UT using two different normalization ranges, the first below 3 km and the second below 4.3 km, over a 0.6 km normalization range. Both the raw calibrated profiles and the water vapor corrected calibrated profiles are shown. In the right panel of Figure 9, the MUSA 1064 nm RCS time series measured during the same time is shown. The aerosol layer observed up to 3.5 km asl is advected from a zonal transport above the Atlantic Ocean and then over Northern-
Central Africa, and likely includes transported mineral dust. Figure 9's left panel comparison clearly reveals that, due to the very low SNR for the CL51 above 3.5 km asl, the molecular calibration is challenging and may result in systematic errors on the retrieved profiles. Aside from the stratocumulus cloud calibration, not addressed in this work, the only possible CL51 normalization to provide a reliable estimation of β' must be performed over a profile of β' retrieved from a reference lidar (like MUSA or PEARL).

CL51 and CS135 dark currents were subtracted from each ceilometer vertical profile to subtract instrumental artefacts affecting the signals, especially in the free troposphere, and to test the feasibility of calibrating ceilometers using the molecular profile. In the CS135, the lack of information in the free troposphere due to data logging problems affected the measured dataset. For the CL51, dark current subtraction significantly reduces the distortions affecting the profiles in the free troposphere. Nevertheless, the ceilometer β' profile calculated for the 5 December 2016 from 17:53 to 19:19
UT (Figure 10), after the dark current subtraction, still has large differences in shape with respect to the PEARL profile, which was successfully calibrated using a molecular profile. The comparison reveals that after dark current subtraction the CL51 β' becomes negative between 2.0 km and 4.5 km asl, indicating that the measured dark currents are inadequate to correct for the signal distortion along the entire profile. This kind of scenario is common throughout the INTERACT-II dataset. The date of 5 December 2016 was chosen because was the closest clear-sky available case to the
date when dark current measurements were taken (22 December 2016).

        It is worth clarifying that a more frequent measurement of dark currents over a longer time window could improve the correction of the signal distortion affecting the ceilometer β'profiles in the free troposphere. Measuring the dark current every 12 hours (once for nighttime and once for daytime measurements), and over a longer integration time, i.e > 1-2 hours, might enable successful application of the molecular calibration. The best practice for performing these
measurements, though primarily of interest to the lidar research community, could be assessed for ceilometers in cooperation with the manufacturers in order to improve dark current correction and allow a more accurate molecular calibration. Tests to assess the value of performing appropriate dark measurements to enable the molecular calibration for the 905-912 nm ceilometers is currently under investigation through analysis of the database collected during the CeiLinEX Campaign (Mattis et al., 2017).

The left panel of Figure 11 shows the profile of the average fractional difference (defined in section 4) between CIAO LIDARs and CL51 values of RCS calculated for 19 cases of simultaneous and collocated measurements, while on the right panel the same is shown for the CS135 but calculated only for 9 cases. The vertical bars again represent the standard deviations of fractional differences. The profiles have been cut off at about 3.5 km asl for both ceilometers because of the low number of available cases with a sufficient high SNR above that altitude. The CL51 underestimates
CIAO LIDARs in the region below 2.0 km asl with a difference lower than 20-30%. It overestimates CIAO LIDARs above 2.0 km, where the decrease of the CL51 SNR with altitude above 3.0 km does not allow the normalization in the FT and the differences with CIAO LIDARs increase to 40-50 %. In the region between 2.0 and 3.0 km asl, where the normalization is applied, the difference is within 10 %. Using the same approach described in section 4 for the MiniMPL, the calculation of the CL51 normalization constant shows a variability within ±46%. While CS135
performances are similar to the CL51 in the region below 3.0 km asl., the difference between CS135 and CIAO LIDARs in the region above 3 km asl vary between ±40 %. The CS135 normalization constant ranges within ±47%.

Figure 12 shows the pdfs of the β' values measured or estimated by CIAO LIDARs and CL51, in the left panel, and by CIAO LIDARs and CS135, in the right panel. The pdfs are limited to β' values below 4 km asl due to the SNR decrease of both the instruments (see above). The intercomparison confirms the agreement between CIAO LIDARs and both ceilometers for the higher values of β', while for lower values, below 0.2-0.3 $10^{-6}$ $m^{-1}sr^{-1}$, the differences are more pronounced due to the lower ceilometers' SNRs.

Finally, Figure 13 shows the scatterplots of 532 nm aerosol extinction coefficient from CIAO LIDARs vs 1064 nm attenuated backscatter from CIAO LIDARs and CL51 in the top panel and from CIAO LIDARs and CS135 in the bottom panel. The scatterplots include just the values measured below 3.5 km asl. For the CL51, differences with CIAO LIDARs in the scatter plot are small and mainly related to the region where β' < 5.0 $10^{-7}$ $m^{-1}$ $sr^{-1}$ and $\alpha_{par}$ > 8.0 $10^{-5}$ $m^{-1}$: in this region, the values observed by CIAO LIDARs correspond to very small values detected by the CL51. For the CS135, though a small number of cases are available, a behavior similar to the CL51 can be identified in the region where β' < 6.0 $10^{-7}$ $m^{-1}$ $sr^{-1}$ and $\alpha_{par}$ > 5.0 $10^{-5}$ $m^{-1}$; these threshold values reveal the slightly better performance of the CL51 when the values of $\alpha_{par}$ are larger for corresponding small values of β' and, therefore, indicates that CL51 has an improved SNR in the night time aerosol residual layer, in particular below 2.0 km asl where the profiles measured by both the ceilometers may be still affected by the correction for the system incomplete overlap.

## 6.  Ceilometer stability

In the previous sections, the overall stability of ceilometers' calibration constant calculated in this paper has been addressed in a statistical sense.  The use of two different multi-wavelength Raman lidars during INTERACT-II did not permit evaluation of the stability of the ceilometer calibration constant in comparison with the lidar system molecular calibration constant, nor did it permit in depth assessment of calibration stability in relation to other parameters (e.g. ambient temperature, aerosol optical depth, etc). Though MUSA and PEARL lidars were compared in the past and may be used almost interchangeably to measure aerosol optical properties, their experimental setups are quite different and therefore different calibration constants are required for the two systems.

Nevertheless, following from the results of INTERACT and in order to assess stability of ceilometer calibration over time, a few tests and studies were performed using the CHM15k as a test-bed. The system (already successfully tested during INTERACT) was not available for much of INTERACT-II due to major maintenance from July to October 2016, therefore it was devoted to this auxiliary testing role taking advantage of the ancillary information provided by the manufacturer through the CHM15k acquisition software. A few tests revealed non-negligible sensitivity of the laser to changes in the ceilometer's enclosure temperature. These changes affect the number of laser pulses emitted per measurement cycle and they are correlated with changes in ambient temperature. To investigate the effect of this behavior on the ceilometer data processing, the whole CHM15k historical dataset available at CIAO was studied. In particular, in Figure 14 the number of laser pulses hourly emitted by the CHM15k is shown as a function of time from 2010 to 2016. The number of plotted points in Figure 14 has been limited anyhow to enable a good visualization. The CHM15k laser specifications provided by the manufacturer are consistent with the measured laser pulse variability, less than <10%. Occasionally, values of the laser pulses' variability up to 15-20 % are also detected.  The specified nominal pulse-to-pulse variance of laser energy is lower than 3%. Interestingly, the laser pulse count variability of 10% does not occur in a random way but, instead, follows a clear dependence on the environmental temperature.  Presumably the ambient temperature affects the ceilometer enclosure temperature, which has the effect of increasing the number of laser pulses in summer and decreasing the number in winter. The number of lasers pulses is included as a multiplying factor in the CHM15k data processing and it is one of the factors contributing the so-called lidar constant within the lidar equation. Presumably, the temperature dependence shown by the laser pulses, likely not unfamiliar to laser experts, directly affects the received signal. The effect is to decrease SNR in cooler temperatures and, therefore, to increase the uncertainty of any calibration method applied to retrieve the aerosol optical properties from the ceilometer data.

This indicates that, across a fixed calibration range (i.e. an aerosol free range to perform the molecular calibration), the normalization constant will range with a behavior similar to that shown by the laser pulses in order to correct for the change in transmitted energy. As a consequence, given that the normalization constant is an operational assessment of the lidar constant plus a residual uncertainty due to the noise, the true lidar constant will have the same seasonal variability as the normalization constant. The reported laser pulses variability can contribute to explain the large variability of the calibration constant (about 58%) calculated during the six-month period of INTERACT (Madonna et

al., 2015) which was only partly due to the variability of MUSA reference lidar (19%). During INTERACT, a direct correlation between the variability of the calibration constant and the seasonal temperature changes was found to be limited ($R^2$=0.6). Nevertheless, the seasonal change in the absolute value of the calibration constant was quite evident and addressed to the coupling of two simultaneous effects (temperature change and decrease in the aerosol loading). The reported seasonal variability of laser pulses also confirms that a calibration constant assessed infrequently will increase the systematic uncertainty contribution. It is possible to estimate over a period longer than 6 months a additional systematic uncertainty in the calibration constant of 10-20 %; over a period of three months the additional uncertainty may reduce to 5-10%. A similar behavior has been observed for the other ceilometers during INTERACT and INTERACT-II, but both the unavailability of single reference lidar during INTERACT-II and the limited database available (only 6 months), did not allow this analysis to be extended to the other ceilometers. It is worth remarking that this seasonal variability has a limited effect on the retrieval of β' for those calibration methods which allow a frequent or continuous calibration (e.g. molecular calibration or indirect calibration using ancillary measurements from a sun photometer). For these methods, the intrinsic accuracy of the calibration method itself is more relevant and can provide the largest uncertainty contribution.

## 7.  Conclusion and outlook

During the INTERACT-II, the newest generation of 905-910 nm ceilometers and a MiniMPL lidar were compared with the CIAO EARLINET multi-wavelength Raman lidars, MUSA and PEARL.

The RCS values measured with MiniMPL and CIAO LIDARs agree within 10-15 % and there are evidences that a re-evaluation of the overlap correction applied in the data processing could further reduce the discrepancies. A preliminary evaluation of the new correction function has been done during the campaign, by using the ratio between MUSA and MiniMPL RCS in the cleanest night time simultaneous measurement session available from both lidars. Nevertheless, a more accurate evaluation of the MiniMPL overlap correction function must be carried out by the manufacturer. The stability of the MiniMPL calibration constant during the campaign was within ±29 %.

The CL51 ceilometer showed a much better performance than the previous generation of VAISALA ceilometers. The CL51 appears to have the capability to detect the molecular signal in the free troposphere; therefore, in order to retrieve the aerosol backscattering coefficient, the calibration of the attenuated backscatter using a molecular profile as a reference can be attempted over integration times longer than 1-2 hours, after the subtraction of dark currents. Nevertheless, signal distortions can have a large effect on the molecular calibration even after dark current subtraction. For this reason, normalization to the multi-wavelength Raman lidar measurements has been performed below 3.0 km asl. Stability of the CL51 calibration constant was within ±46 %.

The CS135 showed improvements compared to the prototype tested during INTERACT.  Its performance was similar to the CL51 in the region below 3.0 km asl (within 20-30% of the CIAO LIDARs attenuated backscatter). However, in the region above 3.0 km asl the differences between the values of the attenuated backscatter are up to ±40 % and molecular calibration is still not feasible for this ceilometer. Stability of the CS135 calibration constant was similar to CL51 and within ±47 %. As already mentioned in the text, it is important to remark that all the statistics on the calibration constants reported in this paper must be used with caution regarding the number of available simultaneous observations for the lidar/ceilometer intercomparison.

Note that both ceilometers were corrected for the effect of the water vapor absorption bands at their operating wavelengths. In addition, it is worth pointing out that the reduced aerosol detection for CL51 and CS135 is also partly due to instrumental processing which is optimized for cloud detection.

Finally, following the primary investigation conducted during INTERACT, a study of the CHM15k historical dataset available at CIAO from 2010 to 2016 has revealed a variability of about 10% for the number of emitted laser pulses which, though within the manufacturer's specification, clearly depends on temperature, with an increase in the number of laser pulses in summer and a decrease in winter. The seasonal behavior shown by the laser pulse numbers directly affects the measured signal with increasing the uncertainty of any calibration method. This contributes to explain the seasonal changes of the CHM15k calibration constant reported during INTERACT (Madonna et al., 2015). The reported seasonal behavior also confirms that ceilometer calibration must be evaluated at minimum every 3-6 months to reduce the uncertainties.

The experience gained during INTERACT and INTERACT-II confirms ceilometers' good performances in the qualitatively monitoring of aerosols in the boundary layer, with enhanced profiling capabilities in the free troposphere only for the most advanced models. Nevertheless, the retrieval of aerosol attenuated backscatter (and of any related optical properties) appears to be affected by instrumental issues which must be improved by the manufacturers in cooperation with the scientific community. Therefore, it is possible to argue that, compared to automatic (backscatter) lidars, though more expensive and equipped with higher-level technologies, the capability of ceilometers of filling in the existing observational gaps within the existing lidar networks at the global scale is in continuous growth, but it is still limited.

590    The datasets during INTERACT-II are made available to the users upon request to the authors though the intention is to make to data available also through the ACTRIS data portal.

## 8.   Acknowledgements

This project has received funding from the European Union's Horizon 2020 research and innovation programme under grant agreement No 654109. The contribution to INTERACT-II of SigmaSpace Corporation, Vaisala and Campbell Scientific, Ltd with the deployment at CIAO of MiniMPL, CL51 and CS135, respectively.

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

**Table 1**: Specifications of MUSA, PEARL and MiniMPL lidars at 532 nm. All the lidars are operated in the zenith pointing mode. RFOV indicates the half-angle rectangular field of view of the instruments.

| Instrument | Wavelength (nm) | Pulse Energy (µJ) | Repetition Rate (kHz) | Configuration | Laser Divergence (mrad) | RFOV (mrad) | Approx. Full Overlap Height (m) |
|---|---|---|---|---|---|---|---|
| MUSA | 532 | $2.5 \times 10^5$ | 0.02 | Biaxial | 0.3 | 0.5 | 400 |
| PEARL | 532 | $5 \times 10^5$ | 0.05 | Monoaxial | 0.125 | 0.5 | 550 |
| MiniMPL | 532 | 3.5-4 | 2.5 | Monoaxial | 0.04 | 0.24 | 2000 |


**Table 2**: Specifications of MUSA and PEARL at 1064nm, CL51 and CS135. All the instruments are operated in the zenith pointing mode. RFOV indicates again the half-angle rectangular field of view of the instruments.

| Instrument | Wavelength (nm) | Pulse Energy (µJ) | Repetition Rate (kHz) | Configuration | Laser Divergence (mrad) | RFOV (mrad) | Approx. Full Overlap Height (m) |
|---|---|---|---|---|---|---|---|
| MUSA | 1064 | $5.5 \times 10^5$ | 0.02 | Biaxial | 0.3 | 0.5 | 400 |
| PEARL | 1064 | $1.2 \times 10^6$ | 0.05 | Monoaxial | 0.125 | 0.5 | 550 |
| CL51 | 910±10nm | 3 | 6.5 | Advanced single-lens optics | 0.15 x 0.25 | 0.56 | 230 (90 % overlap) |
| CS135 | 912±5nm | 4.8 | 10 | Single split-lens biaxial | 0.35 | 0.75 | 300–400 |


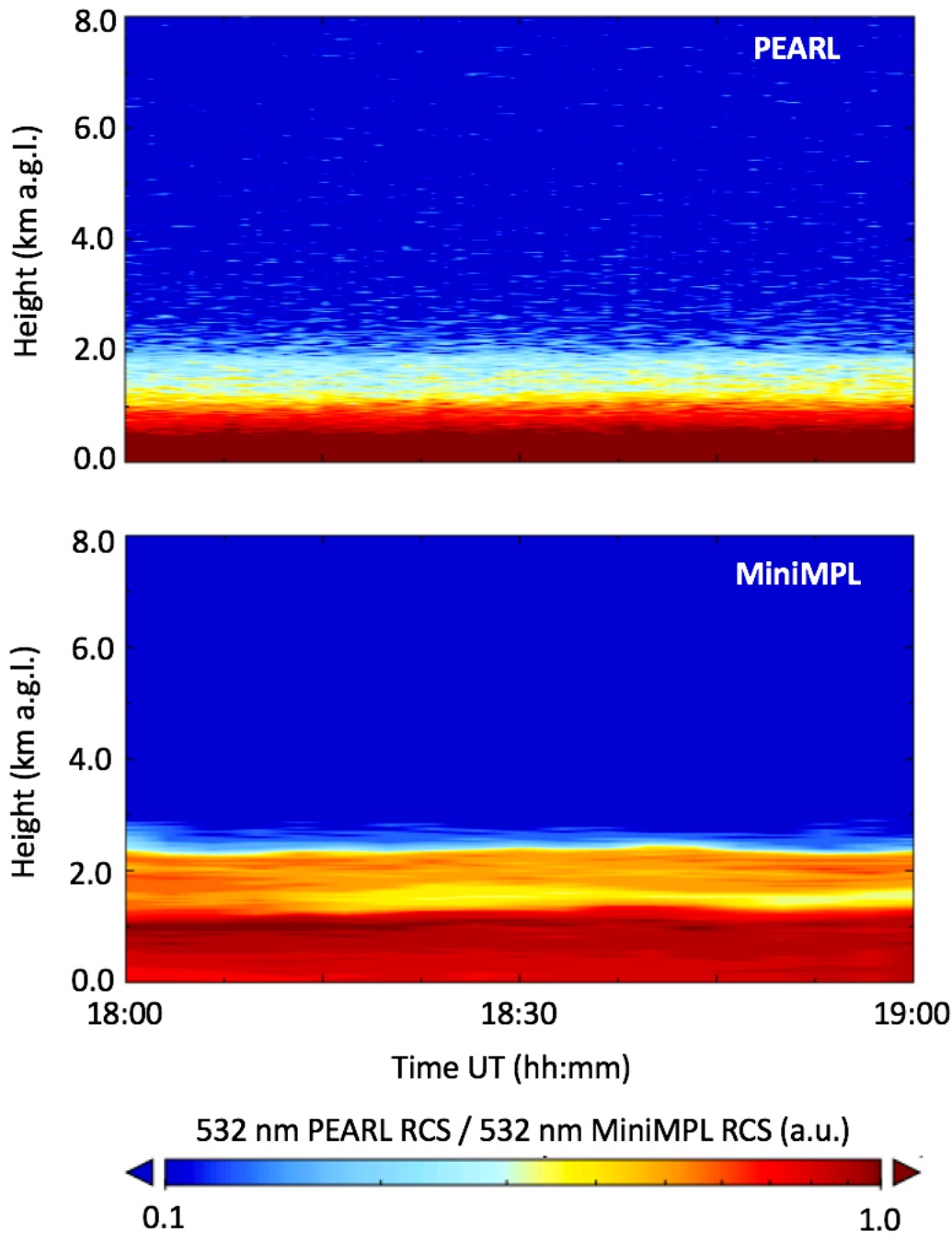

**Figure 1**: Time series of 532 nm Range-Corrected Signal (RCS) measured with PEARL and MiniMPL lidars on 13 October 2016 from 18:00 to 19:00 UT; heights are above ground level (a.g.l.); raw time and vertical resolutions are 1 minute and 15 m for PEARL, and 5 minutes and 30 m for MiniMPL. The color scale shown at the bottom is logarithmic.

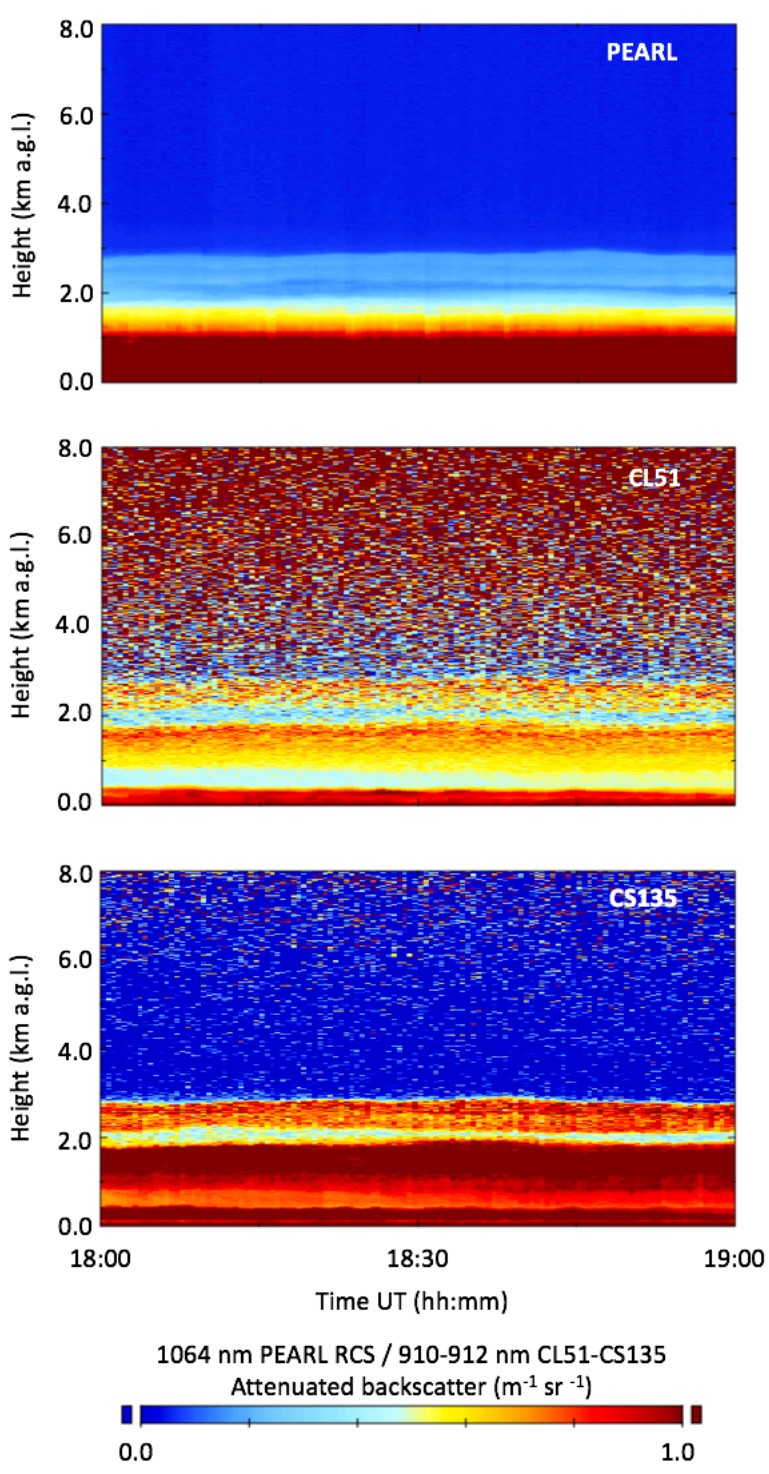


**Figure 2**: Time series of 1064 nm PEARL RCS and of 910-912 nm CL51/CS135 attenuated backscatter profiles as provided through the manufacturer software for the measurements collected on 13 October 2016 from 18:00 to 19:00 UT; heights are above ground level (a.g.l.); raw time and vertical resolutions are 1 minute and 7.5 m for PEARL, 30 seconds and 10 m for CL51 and 30 seconds and 5 m for CS135.


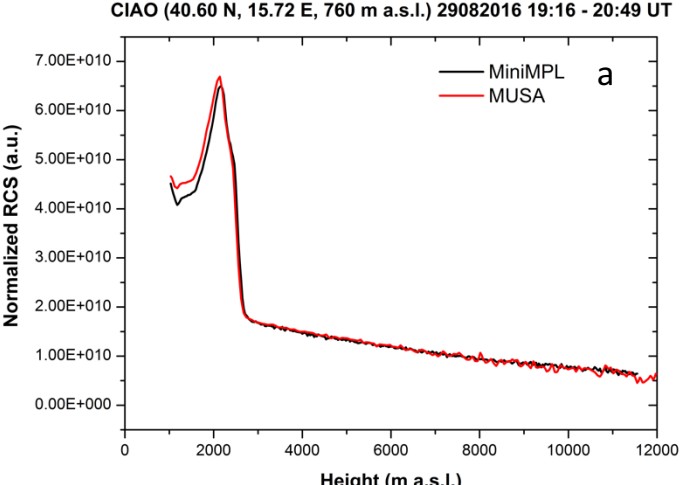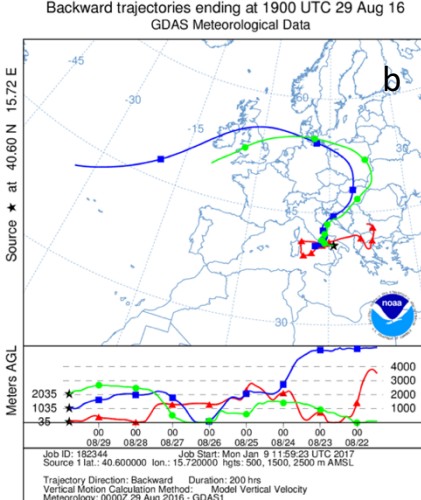

**Figure 3**: In panel *a*, it is shown the comparison between RCS profiles obtained from MUSA and MiniMPL on 29 August 2016 from 19:16 to 20:47 UT; In panel *b*, the corresponding air mass back trajectory analysis performed using NOAA HYSPLIT model is reported; HYSPLIT simulations have been initialized at thethree levels from the ground to the top height of the highest layer observed by both MUSA and MiniMPL lidars.








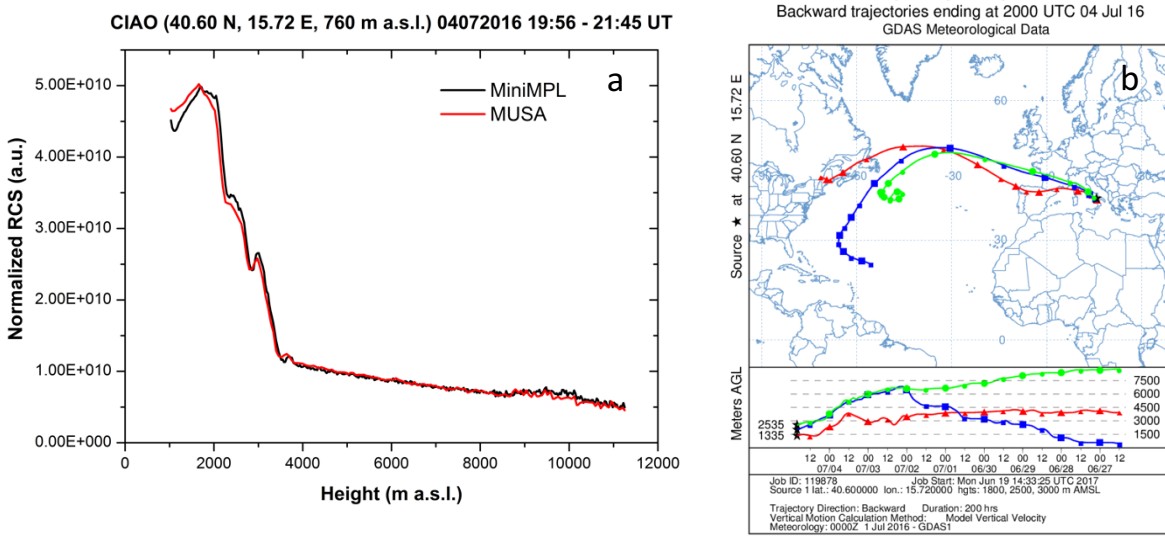


**Figure 4**: Panel *a*, same as Figure 3a obtained from MUSA and MiniMPL on 04 July 2016 from 19:56 to 21:45 UT; panel *b*, same as Figure 3b, the corresponding air mass back trajectory analysis performed using NOAA HYSPLIT model is reported.



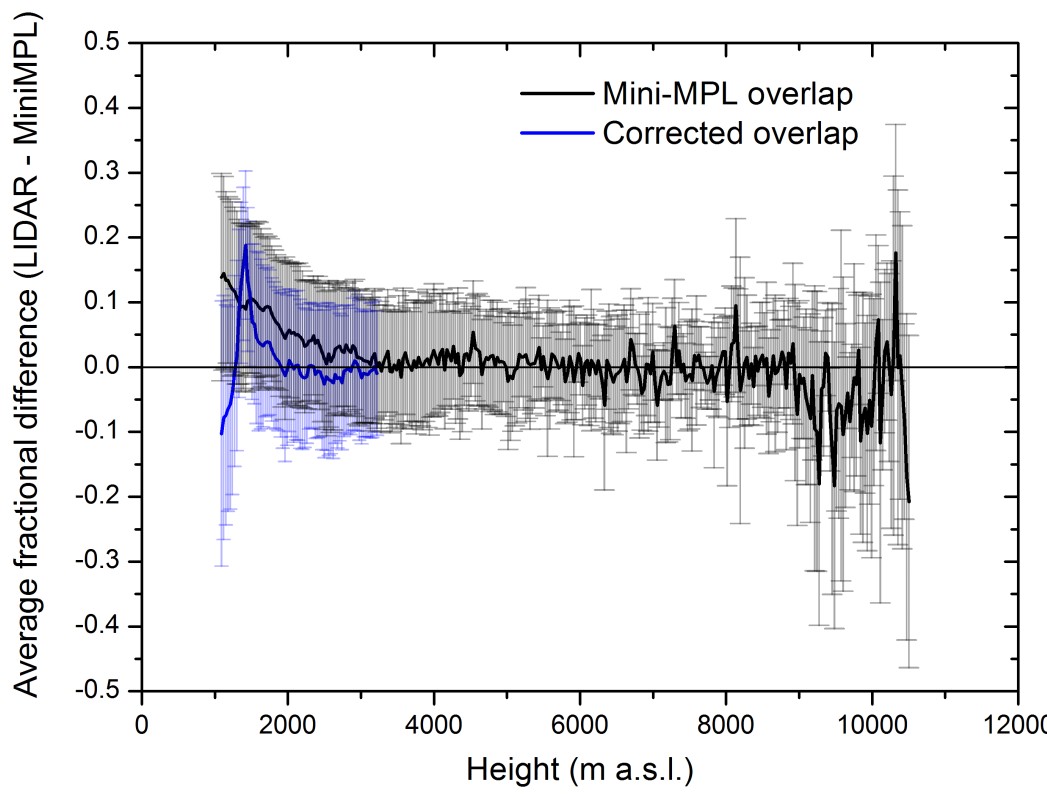

**Figure 5**: Profiles of the average fractional difference between MUSA and MiniMPL values of RCS calculated on 12 cases of simultaneous and collocated measurements (black line). Blue line is the same as black line but applying an additional overlap correction factor to the MiniMPL data processing estimated using the ratio between MUSA and MiniMPL profiles during the cleanest simultaneous measurement session available during INTERACT-II. The vertical bars are the standard deviations of fractional difference.








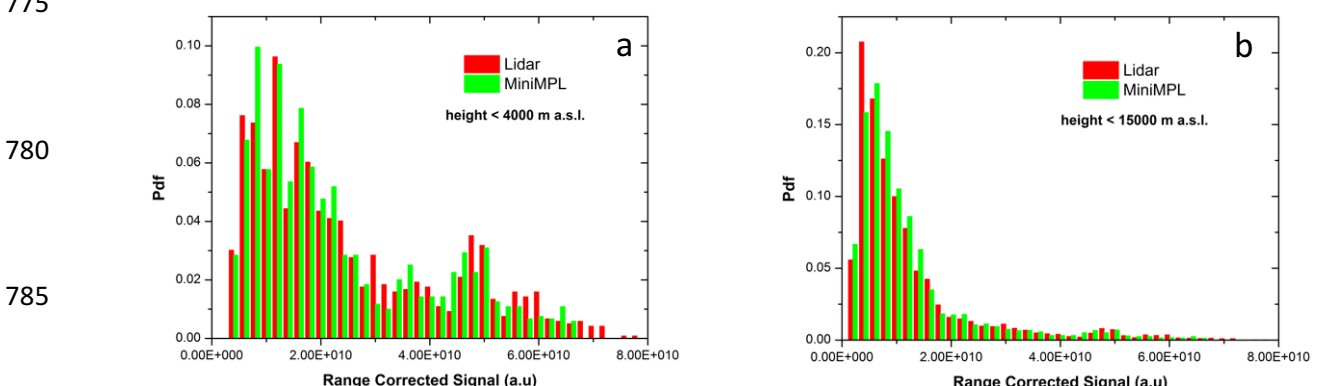



**Figure 6**: Panel *a*, pdfs of the RCS values measured by CIAO LIDARs and MiniMPL below 4 km; panel *b*, same as panel *a* but for the
entire vertical range of observed lidar profiles, below 15 km .




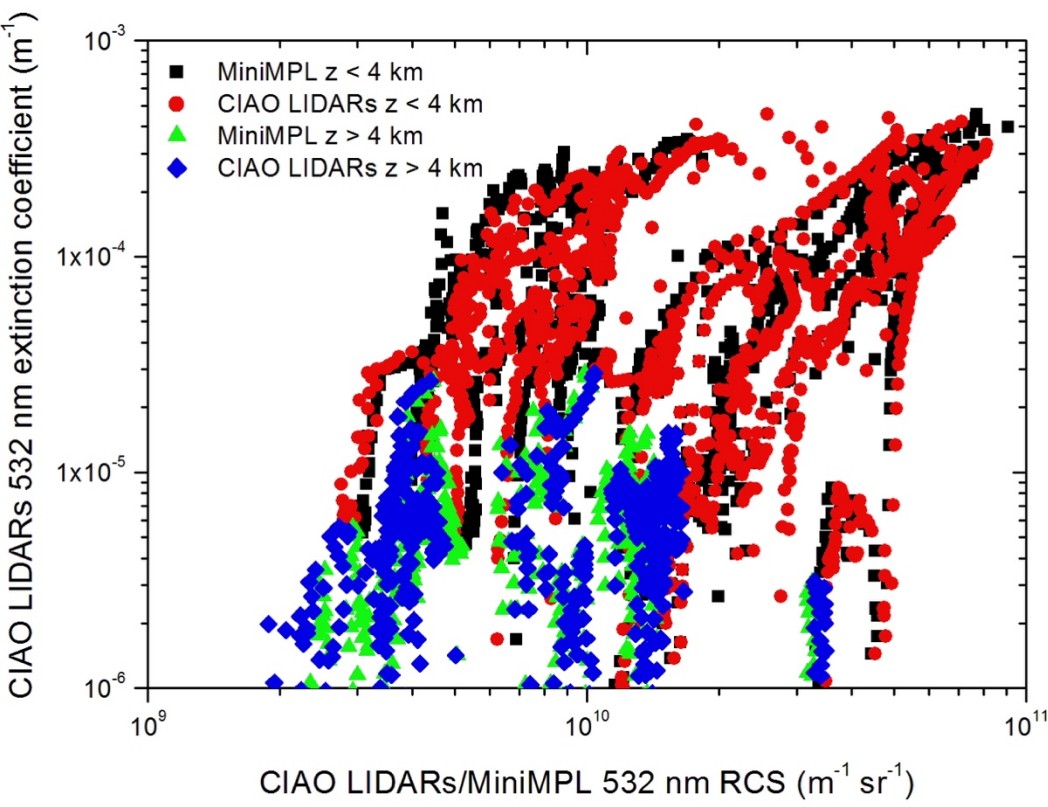

**Figure 7**: Comparison of the scatterplots showing the relationship between CIAO LIDARs 532 nm aerosol extinction coefficient and MiniMPL and CIAO LIDARs 532 nm RCS. Black squares are the values of MiniMPL measured below 4 km, green triangles are the values of MiniMPL measured above 4 km, red squares are the values of CIAO LIDARs measured below 4 km, and blue diamonds are the values of CIAO LIDARs measured above 4 km.

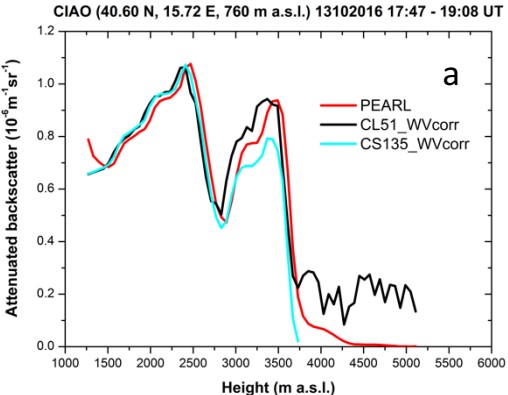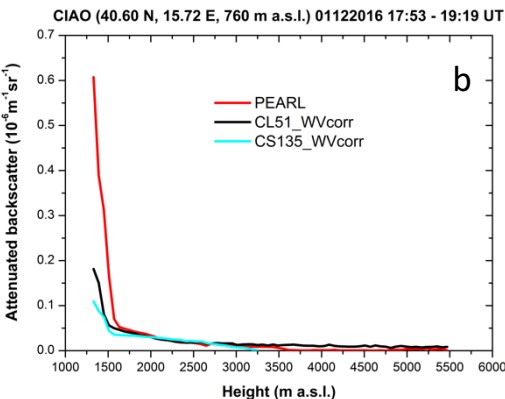

**Figure 8**: Panel *a*, comparison between the attenuated backscatter profiles retrieved from PEARL, CL51 and CS135 on 13 October 2016 in the time interval from 17:47 to 19:08 UT and obtained normalizing the ceilometer profiles on the PEARL profile in the region between 1.8 and 3.0 km; panel *b*, same as panel a, but for the 01 December 2016 in the time interval from 17:53 to 19:19 UT. All the ceilometer profiles are corrected for the water vapor absorption affecting the signal extinction at 910-912 nm.







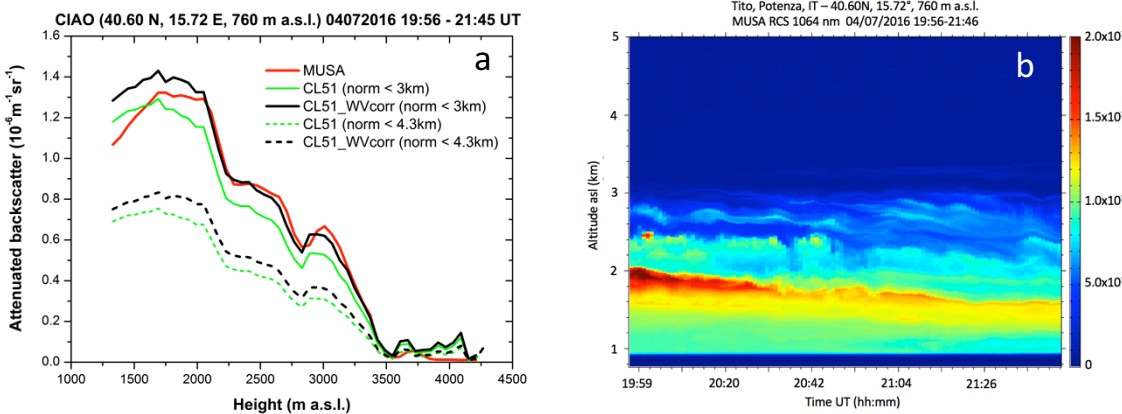

**Figure 9**: Panel *a*, comparison between the attenuated backscatter vertical profiles retrieved from MUSA and CL51 on 4 July 2016 from 19:56 to 21:45 UT and obtained using two different normalization ranges, the first below 3 km (solid lines) and the second below 4.3 km (dashed lines); both the raw calibrated profiles and the water vapor calibrated corrected profiles are shown; panel *b*, time series of the RCS measured with MUSA at 1064 nm during the same time period used to create the average profiles in the panel *a*.

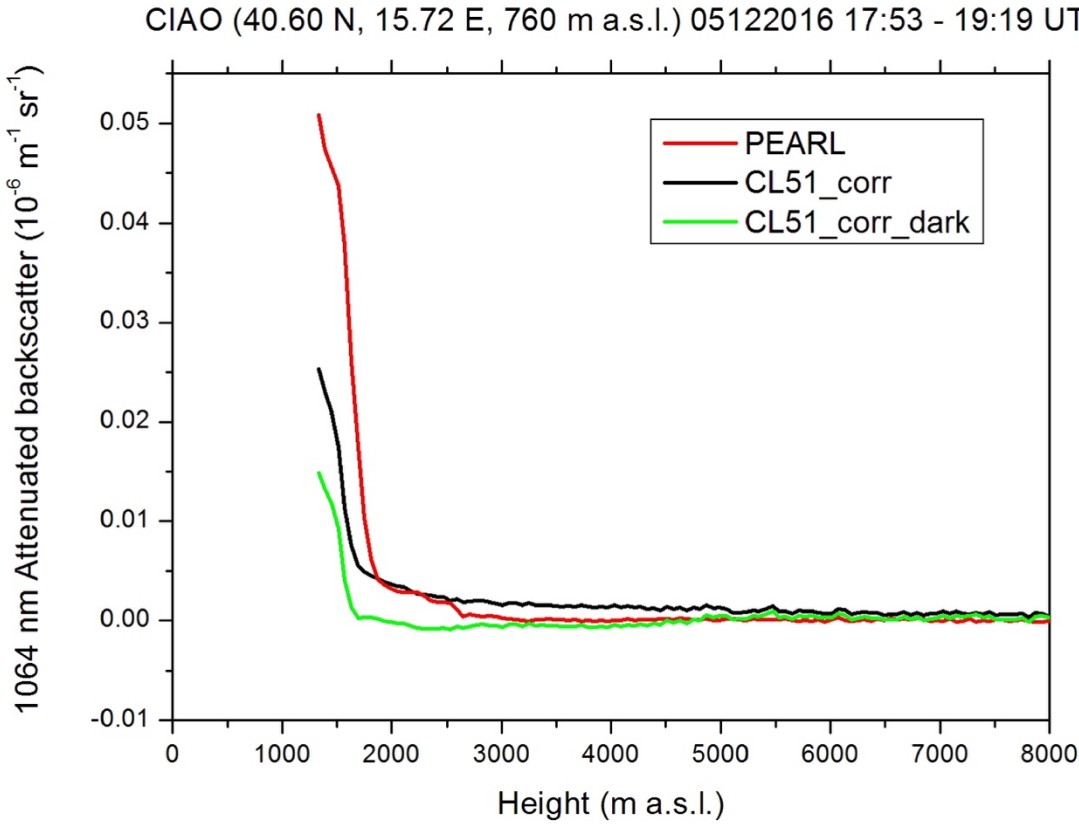

**Figure 10**: Comparison among the attenuated backscatter profile retrieved from PEARL (red), from CL51 accounting for the water vapor absorption at its operating wavelength (dark) and from CL51 subtracting the dark current measured separately and then accounting for the water vapor absorption (blue) on 1 December 2016 in the time interval from 17:53 to 19:19 UT.





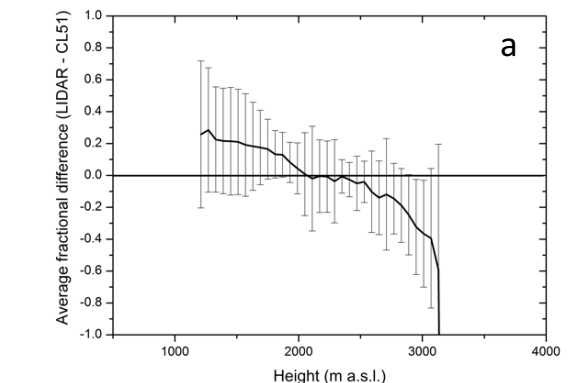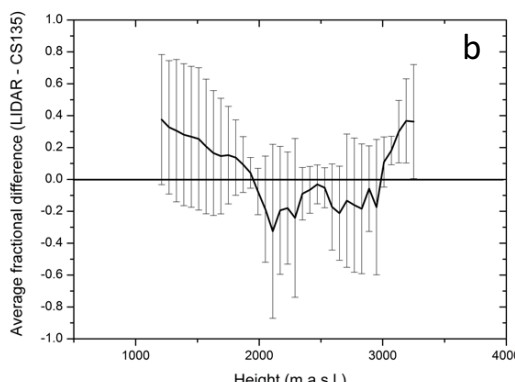

**Figure 11**: Panel *a*, profiles of the average fractional difference between CIAO LIDARs and CL51 values of the attenuated backscatter calculated for 19 cases of simultaneous and collocated measurements; panel *b*, same as panel *a* but for CIAO LIDARs and CS135 calculated for 9 cases. The vertical bars are the standard deviations of fractional differences.

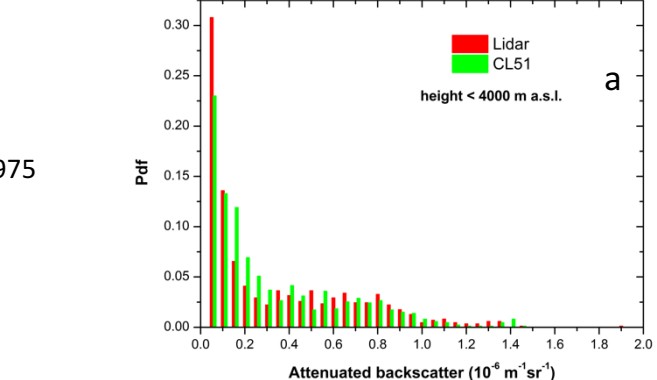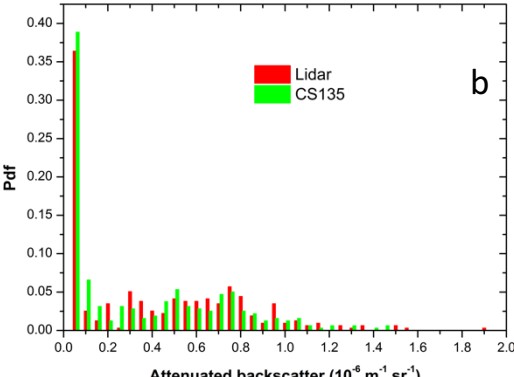


Figure 12: Pdfs of the attenuated backscatter values measured or estimated by CIAO LIDARs and CL51 (panel *a*) and by CIAO LIDARs and CS135 (panel *b*) below 4 km, respectively.






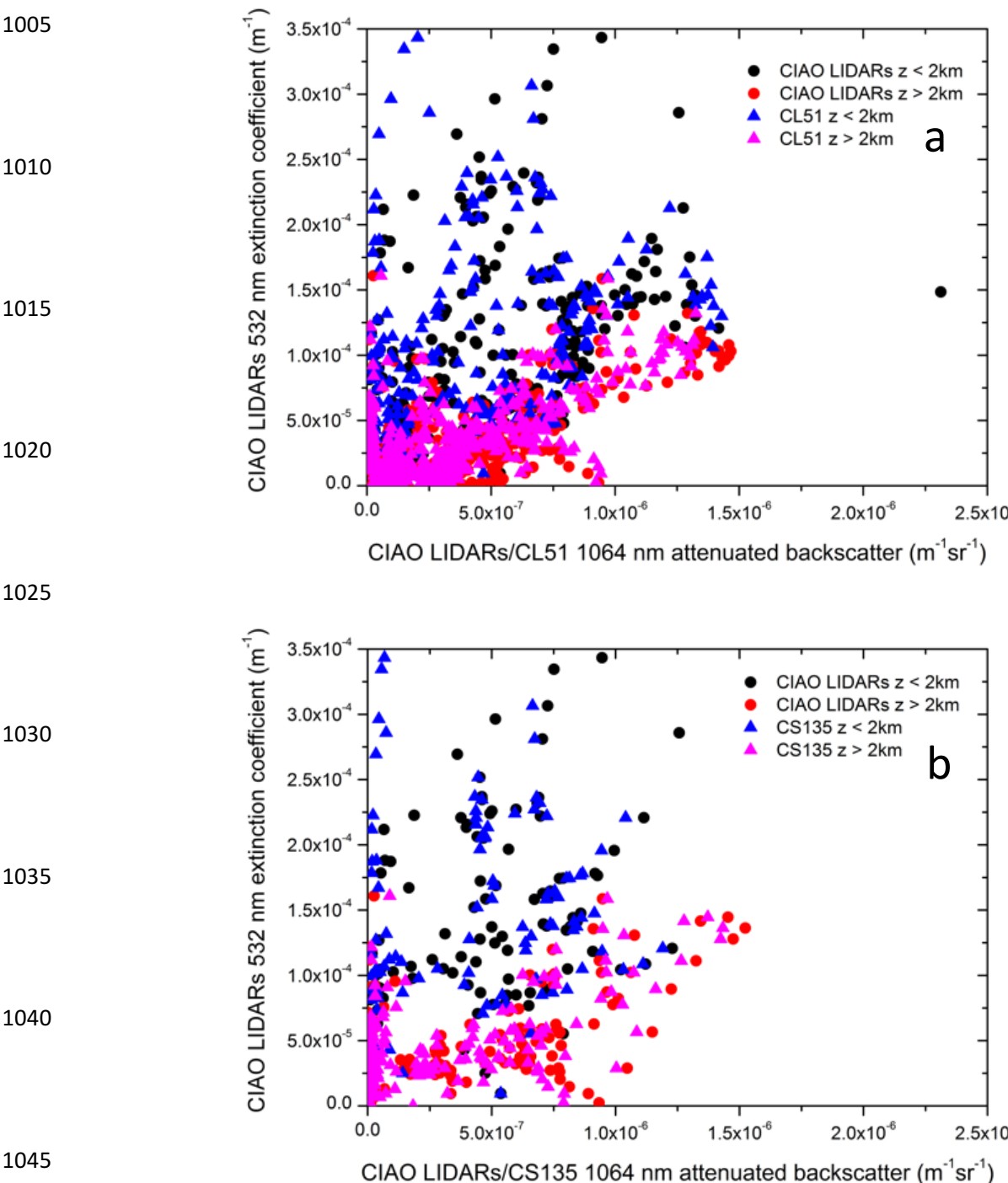

**Figure 13**: Comparison of the scatterplots showing the 532 nm CIAO LIDAR aerosol extinction coefficient vs 1064 nm attenuated backscatter from CIAO LIDARs and CL51 (panel *a*), and from CIAO LIDARs and CS135 (panel *b*). Black dots are the values of CIAO LIDARs measured below 2 km, red dots are the values of CIAO LIDARs measured above 2 km, blue triangles are the values of CL51/CS135 measured below 2 km, and pink triangles are the values of CL51/CS135 measured above 2 km.

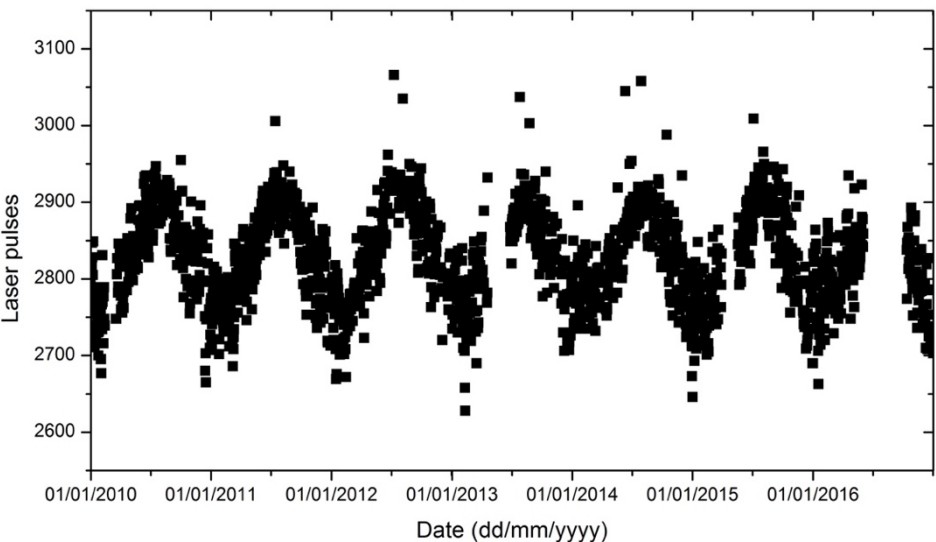

1060    **Figure 14**: Number of laser pulses hourly emitted by the CHM15k as a function of the time for the measurement period from 2010 to 2016.