# Peer review of "Intercomparison of aerosol measurements performed with multiwavelength Raman lidars, automatic lidars and ceilometers in the framework of INTERACT-II campaign"

_Atmospheric Measurement Techniques, 2017_

## Referee Comment (RC1) · Anonymous Referee #2 · 13 Dec 2017

**1   General comments:**

The paper describes an assessment of the performance of a miniMPL and two ceilometers using collocated Raman lidar measurements during the INTERACT II field campaign. This reviewer greatly appreciates the paper's clear organization and good writing, which made it relatively easy to read and review. Another strength is that the discussion is realistic and straightforward about the findings although the findings are not all positive. I think it is an important part of good research to straightforwardly

describe both positive and negative findings and I commend the authors for not over-reaching in their motivational discussion or "spinning" their conclusions to sound more positive than what's justified.

On the negative side, some of the graphs don't seem well designed to answer the questions being asked, and consequently some of the interpretations of the results appear somewhat off-base. There is at least one remaining error in data labeling, and some details need to be explained better. These should be addressed in revision.

The introduction gives a bullet list of the objectives of the campaign but not the objectives of the paper. It would be good to be explicit whether the intention is to address all of these objectives in the paper or only some of them.

Then, please be sure to address those specific objectives in the paper's conclusions also.

Although I was somewhat uncertain about the intended objectives for this paper, some of the objectives listed in the bullet list are addressed rather superficially or even incorrectly. For example, "assess the signal to noise ratio and dynamic range". I don't see analysis specifically addressing these, although some of the analysis of the figures includes some confusion about SNR (see specific comments below). Similarly, "assess the ceilometers' calibration stability and accuracy". There is some discussion about the relatively poor accuracy and about the stability of the lidar instrument itself, but there appears to be confusion in this paper about how to assess stability of the calibration.

**2 Specific comments:**

44. I'm not sure this link is an adequate reference. It links to a pdf of documentation of a piece of software for getting data but doesn't say where to get it. Maybe instead use the link that allows users to get the data–

https://www.dwd.de/EN/research/projects/ceilomap/ceilomap_node.html

47. The authors criticize the lack of global lidar coverage and lack of homogeneity within current lidar networks as if this is a motivator of the current work, but it isn't clear how the current work advances the goal of homogeneous lidar coverage. I do understand (from the next paragraph in the manuscript) and agree that vetting cheaper lidars might enable more global coverage, but it doesn't follow that such a network with a wide range of lidar capabilities will be more homogeneous than existing networks. Better to delete the sentence starting "Even when federated" or put in more discussion making the motivation and link to this work more clear.

58. "have been already investigated". That's fine, but in the next sentence or soon thereafter, you need to explain what the new contribution of this paper is.

62. "retrieval . . . can be performed using the molecular backscattering profile". Please be more specific. You mean the calibration can be performed using the molecular backscattering profile in a region where there is negligible aerosol. Without these additions, the phrase "the retrieval can be performed using the molecular backscatter profile" sounds like the Raman or HSRL retrieval.

215. Each lidar instrument has its own version of the quantity being considered, all with different names: attenuated backscatter, range-corrected signal, and normalized relative backscatter. It's a little confusing, but with some effort I see why you made these choices. It would be helpful to have a paragraph (earlier than this) where all three quantities are described in one place and the reasons for using different quantities for each instrument are provided.

275. Please explain the normalization further. Is the MiniMPL normalized to match PEARL in the normalization region on a profile-by-profile basis for every profile?

322. If you mention after-pulse correction as a possibility, I think it needs to be supported. Otherwise it just sounds random and speculative.

325. Are the 12 cases all the measurements available from the whole six month deployment period, or have these been selected from a larger dataset? (Were the lidars operating continuously?)

325. RCS or normalized relative backscatter? I thought that RCS meant non-normalized signals, so they could not be compared between two instruments? If there's no useful distinction in the terminology, it would be better to just use one name instead of three.

338. "The good stability of the MiniMPL calibration ... is shown by the small variability (10%) of differences in the normalization region". I have a major problem with this statement. This one must be addressed. Aren't the profiles for all 12 cases normalized to the Raman lidar in the normalization region? So, for each profile, the normalization constant is divided out and each profile is independently set to have zero average difference in the normalization region. That means that to assess the profile-to-profile variability in the normalization constant, you'd have to explicitly look at the 12 normalization constants. That information is not present in the data shown in Figure 5. Variability in the normalization region in Fig 5 is only representative of high-frequency noise within the normalization region, so it informs you about the precision, but not about the stability over time.

Figure 7. The differences in the scatter plots are very hard to make out given the data being compared are in two different plots. A figure showing both in the same figure would be better. For example, consider making a scatter plot of CL135 vs. MUSA/PEARL attenuated backscatter directly, and color code by extinction or stratify different ranges of extinction into multiple sub-plots. Accompanying them with another set color coded or stratified by altitude would also be helpful, I think, given that the interpretations of this figure in the text are related to specific altitude regions (the overlap region and the free troposphere). (Same comment for Figure 13.)

348. You say that the choice of aerosol extinction for the y-axis of Figure 7 (and 13)

is to reveal differences in sensitivity to different aerosol types. In fact, you have not mentioned different aerosol types in your interpretation at all. Indeed this task would be quite difficult with the information given in the figures since the relationship between attenuated backscatter and aerosol extinction is related not just to lidar ratio (indicator of aerosol type) but also to the amount of attenuation, and the attenuation may be a more dominant effect in this data set. If you really wanted to distinguish different aerosol types, you might consider including lidar ratio (from the Raman lidar) in the analysis. If you don't care about aerosol type, then probably just delete the statement about them.

354. "The most evident differences between the two lidars can be identified for values of extinction larger than about $5.0x10^{-5}m^{-1}$ where miniMPL shows a broader scatter." It's very difficult to see this. I see that miniMPL has a bit more data at the low end of the x-axis and MUSA has a bit more data at the high end of the x-axis, but it is by no means obvious.

355. "Described above" What does this refer to, the unexplained statement about after-pulse correction?

371. SNR. There seems to be some confusion between signal level and signal-to-noise ratio. The text says the CS135 SNR decreases above 3500 and the CS 51 SNR is higher. To me, it looks like the signal of CS135 decreases and the signal of CS51 is higher, but the noise in the CS51 signal is also quite high and it clearly does not agree with the more reliable Raman lidar, so it's likely this higher signal is an artifact. Your graph doesn't show SNR explicitly enough to aid in analyzing the SNR. I think you would have to look at both signal and noise and analyze noise levels explicitly to be able to make quantitative statements about how the SNR for the two instruments compare. From figure 8 I think you can say "The CS135 signal strongly decreases . . .. The CL51 signal is higher but the noise suggests that it is not reliable to detect the residual aerosol. . ."

Figure 10. Please check the units. Is the exponent -6? Or -5? Compare to Figure 7 which I think should be the same PEARL profile.

422. Similar to my comment at line 338, I don't agree with this. Is each case normalized separately? If so, then the variability of the normalization constant is not represented in this plot. The error bars are related to the amount of variability over the few hundred meters of normalization range, but not to the stability of the normalization constant over time.

425. "The standard deviation of the normalization constant." Is this calculated separately by keeping track of the individual profile normalization constants? That's the correct way to do it.

439. "better performance of the CL51 when the values of extinction are larger for corresponding small values of backscatter and therefore indicates its improved SNR in the FT". Is it really true that these large values of extinction with corresponding small values of backscatter are in the free troposphere? Wouldn't small backscatter values in the free troposphere more likely be accompanied by small values of extinction? It seems more logical that if there are small backscatter values when the extinction is large, that means there is significant attenuation, so the points are more likely low in the atmosphere below significant aerosol layers. This is important to check and clarify since you seem to be drawing a major conclusion (better performance of CL51 in the FT) almost wholly from this subtle and hard-to-interpret pattern.

443. "The overall stability of ceilometers' calibration constant . . .has been addressed in a statistical sense." I don't see any analysis of the overall stability of the calibration constant, see comments at line 338 and 422.

464. "in general is embedded" – please be more specific. Do you mean "is directly proportional to"? If so, please say that. If the relationship is more complicated than that, please include the equation.

470. "the calculated embedded constant". Does this mean lidar constant? Please say that. I think it would be good to put in some clarification that the calibration constant is an operational assessment of the lidar constant (which may have some noise or error). So I think what you're saying is that if the true lidar constant has seasonal variability but a calibration constant is only assessed infrequently, then there will be a systematic error in the calibration constant.

471. "what was reported during INTERACT". What was reported? Be more specific.

471. "This partly explains". To me this finding of a temperature dependence suggests a hypothesis, but I don't see any testing or exploration of the hypothesis. Is there any indication that the variability during INTERACT was correlated with temperature? (I see in the earlier paper it was believed that there was, but there was no quantification of the correlation, and that information is missing entirely from this paper).

471. As your continuing discussion points out, it doesn't seem that the size of the effect matches well at all. If the lidar constant is linearly related to the number of laser pulses, then the variabilities are also linearly related, and so 10% variability in pulse count can hardly explain 58% variability in the calibration constant. I think maybe it would be best to change the wording to remove or further deemphasize the "This partly explains" clause. While I agree that you have demonstrated that operators must be aware of temperature as a source of variability, as an investigation of the cause of the observed variability in the INTERACT observations, this is inconclusive at best.

486. "most of the difference could be reduced after a reevaluation of the overlap correction". This statement in the conclusions is quite a bit stronger than the statement in the body of the text. In the text you demonstrated that reduction of the error was possible for a single case when the Raman lidar is available to show the true shape of the overlap region, but that it couldn't be corrected in most cases.

492. "The CL51 is able to detect the molecular signal in the free troposphere". I'm not convinced this was demonstrated.

500. Since the introduction suggested a main motivation was "to understand to what extent automatic lidars and ceilometers are able to provide an estimation of the aerosol geometric and optical properties and fill in the geographical gaps of the existing advanced lidar network", it would be good to see some conclusion about this question here. You have said earlier "the only possible CL51 normalization to provide a reliable estimate of attenuated backscatter profile must be performed over a profile of attenuated backscatter from a reference lidar (like MUSA or PEARL)." These seems to argue against the usefulness of ceilometers for filling in existing gaps. Whether or not I am correctly guessing your conclusion, some discussion belongs in the conclusion section.

**3  Technical & grammatical:**

143. Is it 16 optical channels? The description in the following sentences seems to say 16, not 17. Is something left out or is there a typo, maybe?

231. Probably "temperature" rather than "thermostat". A thermostat regulates temperature.

235. Instead of using "beta", spell out attenuated backscatter or use the symbol $\beta'$ that was already introduced.

344, 353, 354, elsewhere? Fix formatting of numbers in scientific notation

367. Possible missing word "between" 2.5 and 3.5 km asl

416. Delete the word "average"? I think you probably are reporting the standard deviations of the fractional differences, not the standard error of the mean. If you are reporting the standard error of the mean, please use that terminology rather than "standard deviation of the average".

448. Replace indifferently with interchangeably

451. "over the time", delete "the"

504. "INTERACT-II". Should this be "INTERACT-I"?

Figure 1. A log scale might be more informative for this quantity.

Figures 3, 4, 6, 8 . The label "LIDAR" should be "MUSA", "PEARL" or "MUSA/PEARL"

Figure 7. the axis labels are really small and it's not possible to zoom them in enough to make them clear. It would be good to remake these with bigger axis labels. (But see above: I also have a suggestion for a different plot style altogether.)

Figure 7 caption. Please state the time & date of the comparisons.

Figure 8. "Using t[w]o normalization ranges (below 3 km and above 8 km)". It appears that this is incorrectly pasted from another figure. Figure 8 doesn't seem to have two normalization regions.

Figure 11 caption, line 841. "standard deviations of the fractional differences" not "average", I think (see above)

Figure 12 caption. "and along the whole observed atmospheric column". It appears this is a copy-paste from Figure 6 and should be deleted.

---

## Referee Comment (RC2) · Anonymous Referee #1 · 21 Dec 2017

AMTD: Intercomparison of aerosol measurements performed with multi-wavelength Raman lidars, automatic lidars and ceilometers in the frame of INTERACT-II campaign

General Comments

The paper "Intercomparison of aerosol measurements performed with multi-wavelength Raman lidars, automatic lidars and ceilometers in the frame of INTERACT-II campaign" reports the results of a campaign using a variety of instruments to measure aerosol in cloud-free or clear-sky conditions. While the authors report interesting

results, I think that they could make the analysis more rigorous and motivate the work more clearly. I have made recommendations below.

Specific Comments

1. I recommend that the authors provide general motivation in the introduction for this study. Why does anyone need to measure atmospheric aerosols using these types of instruments? Why is this intercomparison needed? Is it to help design better networks for measuring pollution, for example? I would like to understand this and to make sure the audience understands how the intercomparison gives us important and useful information. Can the authors say anything specific about the aerosols that were measured (type or other properties) during the campaign?

2. Please can the authors explain, again in a general way, which of the instruments is expected to measure aerosols (of a given type) most accurately and why. For example, can you give a general sense of where (in the atmospheric column) the instruments are expected to give the best results? And why? Perhaps it would be helpful to touch on differences in wavelength here as well as other differences in hardware or firmware? I realise that none of the instruments gives us "truth", but can the authors give the reader a sense of the accuracy expected? Thus, when the differences are reported, the readers immediately understand which of the instruments is believed to be closer to the true observed quantity.

I suggest these two points in order to give the reader a better sense of why these particular instruments are important to study (as I think that they are) and to make a stronger case for why the intercomparison analysis in this paper matters to the community.

3. In section 5 MUSA is referred to as the reference signal in the full overlap region. Why is MUSA the reference? Is it expected to be the highest standard of measurement to which we want to compute the ceilometer observations?

4. Please define the "fractional difference". For example in section 4 Paragraph 5,

"average fractional difference" is not defined and later in the paragraph (line 327) an "average difference" is increasing. Are these the same metric? The authors need to define clearly the measure or measures of difference applied to the results.

5. There are a few places where the authors discuss "random uncertainty" (section 4 for example in line 322). Please could the authors define how they determine the random uncertainty? Also, if there are some statistical tests being performed to assess differences then please state which tests are being used. For example, is there a null hypothesis of random white noise?

6. At the end of the technical corrections, I have placed a number of comments on the figures which need to be addressed.

Technical Corrections

Title: Please change "frame" to "framework".

Text:

1. Many acronyms are undefined in the main body of the paper. To aid the reader, please explicitly define the following: CNR-IMAA, EARLINET, FOV, FWHM, GRUAN, RAOB, HYSPLIT and APD in line 149 (is it Avalanche Photo Diode?) used before line 161 Avalanche Photo Detector are these the same "APD"?,

2. Please put units on the RCS. I believe that the authors are using "arbitrary units" (a.u.) throughout. Is this correct? Can a.u. be placed next to all the measurements please?

Line 22 Is average difference a root mean squared difference? Absolute difference? Or something else?

Line 29 Rewrite to something more like: "Some tests performed during this campaign using the CHM15k ceilometer made it clear that the CHM15k historical dataset (2010-2016) available at CIAO should be reviewed in order to evaluate the potential effect of

..."

L 39 systems not system

L 123 change to "...unit consists of a Cassegrain..."

L 135 change "system was operative" to "system operated"

L 137 change "when MUSA moves" to "when MUSA was moved"

L 167 remove "is". change to "The Vaisala ceilometer CL51, the second generation..."

L 169 add: "to diagnose vertical visibility"

L172 at "the" surface

L 175 remove is. "The instrument used in INTERACT-II..."

L184 add punctuation: " troposphere; thus, limiting"

L188 "in principle, but are eye-safe..."

L192-193 "limits which permit these ceilometers to operated unattended."

L 194 Change to "and, therefore, enhanced performance" not plural performances

L213 CHange to "In contrast to the ceilometers,"

L215 Split up sentence: "...performance. Using RCS allows a comparison to be made between the ..."

218 I am confused about what covers a shorter vertical range? Perhaps split into 2 shorter sentences here and be more explicit.

L220 assumption should be plural.

L221 What are you comparing here? I would say: "To perform the comparison between the MiniMPL and the MUSA/Pearl,"

L223 say "...identified as negligible qualitatively using quicklooks..."

L226 change to "...applied to the data, but systems..."

L227 "...interpolated to the MUSA/PEARL..."

L237 Can you please clarify what is more frequent in the FT?

L239 Please could you briefly (in a sentence) say why the assumption of < 1% is a good one? I can see there is a reference, but a quick explanation would be helpful, if a brief one is possible.

L243 "... consider the attenuation of the backscatter signal by water vapour. In this study, a method for correcting for the attenuation by water vapour is..."

L260 "Drop "a". "to avoid unpredictable"

L271 Drop "for". "calculated through"

L275 remove the convoluted phrase at the beginning. Just write: "A comparison of..."

L276 What does "their own time" mean? Is it at the time resolution of the measurement taken by each instrument?

L280 "for MiniMPL the time resolution is 5 minutes and the vertical resolution is ..."

L282-283 "operator routinely checked each instrument during INTERACT-II to ensure that each one was performing according the the manufacturer specifications."

L285 "of each instrument"

L286 "weekly check on each instruments' acquisition..."

L290 "flooding method. Additionally, specific treatments to remove the stronger dust spots were performed..."

L293 "measurements were made twice..."

L295 "profiles before normalization using the lidar..."

L295 "...dark current measurements were routinely..."

L299 "Simultaneous observations of aerosol made using the multi-wavelength..."

L305-306 are very confusing. Model levels are used under some kind of observed threshold, but the "top layer" is very confusing. What is being done?

L305 Need a reference for the NOAA HYSPLIT model. A paper or a technical report should be cited when the model is first discussed.

L309 and L318 Just say "good agreement" remove "a very good"

L310 What is the RCS random uncertainty?

Section 4 paragraph 4 does not state in the text which data are being compared.

Section 4 Paragraph 5 is confusing because fractional difference and difference are both being used without being defined.

L 344. Number is not in scientific notation.

L 348. Please define the variable alpha_par in the text explicitly. It sounds like alpha is the aerosol extinction coefficient, but "par" is undefined.

L 352 Is the output profile from Raman PEARL lidar? If so, is it interpolated to the same resolution as the RCS from which instrument?

L 353-356 Numbers are not type set correctly, missing the multiplication sign.

L364 What are these different altitude levels?

L367 "...CS135 in the region between 2.5 and ..."

L370 ranges should not be plural

L377-378 "..because MUSA is considered the reference signal only in the full overlap...." Has this been stated before? Has MUSA been the reference all along?

L394 I would suggest the wording should be changed to "Dark current measurements or profiles", not just "dark currents"

L403-404. I wouldn't talk about the chosen data being "not random" as it suggests some kind of statistical test, but instead say something like: "The date 5 December 2016 was chosen because it was the closest clear-sky case to the date when the dark current measurement was taken (22 Dec 2016)." or something more concise and descriptive.

L421 increases should be increase

L423 constant is misspelled.

L423 change to " The CS135 measurements are similar to the CL51 in the region..."

L424 Start a new sentence with the comparison stated explicitly. Is it like this? "While in the region... the differences between the CS135 and the MUSA/PEARL vary between -40% and +40%."

L436. What is the effect that is similar between the two ceilometers in the region given by range of attenuated backscatter and extinction?

L439 which ceilometer "its" referring to? Which ceilometer shows improvement in the RCS?

L448 I don't think that indifferently is the correct word to use here. Do you mean interchangeably?

L453 Change "to the ancillary" to "of the ancillary"

L454 change to "revealed non-negligible sensitivity"

L460 "values up to 40-50%" The percentages in this paragraph are given without a reference value. Which variable are we discussing? 40-50% of what reference value?

L462 Punctuation needed: "Presumably,"

L462 affect should be plural: affects

L463 remove "a" from "a decreasing"

L464 Change to "The number of laser pulses is included as..." The word assimilated might confuse readers as it is generally used in the context of data assimilation.

L466 change to "...signal. The effect is to decrease SNR in cooler temperatures... increase the uncertainty....from the ceilometer data

L469 It is not clear what "This" means. Please state explicitly.

L469 "aerosol free range" may be clearer to change to "aerosol-free atmosphere" or "aerosol-free atmospheric column"

L470 What exactly is the "calculated embedded constant"?

L 475 Just say "Similar behaviour..."

L479 allows should be singular

L480. Split the sentence. "...photometer). For these methods..."

L484 "Say "During the INTERACT-II campaign"

L502-503 Change to "it is worth pointing out" and "partly doe to the instrumental processing that is mainly.."

L512 "made available to the users"

L512-513. Split this sentence into 2.

Figures:

1. All sub-panels within all figures should be labelled with letters a,b,c, etc. 2. In the text and captions all of the sub-panels in the figures should be referred to using the

figure number and letter together. Please do not use left/right, top/bottom. The letters make the text concise and precise. For example caption for figure 8 should read more like: "Panel a shows attenuated backscatter retrieved from ... Similarly, panel b shows the same comparison but for 01 December ..."

Figure 3, 4,6, 8, 9, 12 have a red line (or red bar) labelled "Lidar" but MUSA is in the caption. Lidar is not specific enough. Please make the legend consistent and more precise. Is it MUSA Lidar? In contrast, for example, Figure 10 has a red line called PEARL which is also a Raman lidar like MUSA.

Figure 4. Caption is confusing. Can authors please explain what they mean by "using NOAA HYSPLIT model started at the three levels from the ground the top layer observed by MUSA and MiniMPL lidars"? Are we talking about model levels? What is "the top layer observed"?

Figure 5. Should read "Blue line is the same as the black line but..." Also, captions usually put the line colour or line style in parentheses like this: "Profiles of the average fractional difference (black line)..."

Fig 9 End of caption: " Panel b shows the attenuated backscatter vertical profiles taken using the MUSA/PERAL lidar which operates at wavelength 1064 nm during the same time period as was used to create the average profiles in panel a."

Fig 10 Change to "Comparison between" not among. Also the line colour is "green" not "dark". This is the line with the dark current measurement subtracted away but the line is green.

Fig 11 Change "calculated on" to "calculated for"

Figure 14: What time does each square represent? Can't be 30 s resolution?! There are 7 years on the x-axis. How were the laser pulses averaged?

---

## Author Comment (AC5) · 23 Jan 2018

**Reply to the comments by the anonymous Referee #2**

**The authors gratefully acknowledge the reviewers' effort in improving the quality of the manuscript. Below, all the major, general and specific reviewers' comments are addressed (in bold). The remaining minor comments have been all of them fixed in the new version of the manuscript.**

**1 General comments:**

The paper describes an assessment of the performance of a miniMPL and two ceilometers using collocated Raman lidar measurements during the INTERACT II field campaign. This reviewer greatly appreciates the paper's clear organization and good writing, which made it relatively easy to read and review. Another strength is that the discussion is realistic and straightforward about the findings although the findings are not all positive. I think it is an important part of good research to straightforwardly describe both positive and negative findings and I commend the authors for not over- reaching in their motivational discussion or "spinning" their conclusions to sound more positive than what's justified.

On the negative side, some of the graphs don't seem well designed to answer the questions being asked, and consequently some of the interpretations of the results appear somewhat off-base. There is at least one remaining error in data labeling, and some details need to be explained better. These should be addressed in revision.

The introduction gives a bullet list of the objectives of the campaign but not the objectives of the paper. It would be good to be explicit whether the intention is to address all of these objectives in the paper or only some of them.

Then, please be sure to address those specific objectives in the paper's conclusions also.

Although I was somewhat uncertain about the intended objectives for this paper, some of the objectives listed in the bullet list are addressed rather superficially or even incorrectly. For example, "assess the signal to noise ratio and dynamic range". I don't see analysis specifically addressing these, although some of the analysis of the figures includes some confusion about SNR (see specific comments below). Similarly, "assess the ceilometers' calibration stability and accuracy". There is some discussion about the relatively poor accuracy and about the stability of the lidar instrument itself, but there appears to be confusion in this paper about how to assess stability of the calibration.

**2 Specific comments:**

44. I'm not sure this link is an adequate reference. It links to a pdf of documentation of a piece of software for getting data but doesn't say where to get it. Maybe instead use the link that allows users to get the data– [https://www.dwd.de/EN/research/projects/ceilomap/ceilomap_node.html](https://www.dwd.de/EN/research/projects/ceilomap/ceilomap_node.html)

**The authors have been not able to find a reference, therefore they simply modified the link according to the reviewer's suggestion.**

47. The authors criticize the lack of global lidar coverage and lack of homogeneity within current lidar networks as if this is a motivator of the current work, but it isn't clear how the current work advances the goal of homogeneous lidar coverage. I do understand (from the next paragraph in the manuscript) and agree that vetting cheaper lidars might enable more global coverage, but it doesn't follow that such a network with a wide range of lidar capabilities will be more homogeneous than existing networks. Better to delete the sentence starting "Even when federated" or put in more discussion making the motivation and link to this work more clear.

The sentence indicated by the reviewer #1 states:" Even when federated networks have been set-up by international stakeholders (e.g. GALION – GAW Lidar Observation Network), the different practices adopted within each of the federated networks (e.g. EARLINET, MPLNET, ADNET, LALINET) significantly affect the homogeneity of the collected measurements; at present only one example of a coordinated monitoring of a global scale event (Nabro volcanic eruption) has been provided in literature (Sawamura et al., 2011)."

The authors used this sentence to acknowledge that GALION could improve our understanding of aerosol at the global scale but not without a harmonization effort to spend across the federated network. Clearly this is the same for commercial instruments, though a smaller number of "degrees of freedom" should be involved to achieve a global harmonization of the provided data. Actually there 7-8 commercial instruments (including the old models) which covers the majority of stations available worldwide and equipped with an automatic lidar or a ceilometer.

In contrast, a network like GALION, though based on a smaller number of station should deal with many home-made and commercial instruments together, which surely led to an increased level heterogeneity within the network. EARLINET and MPLnet already spent a lot of effort to increase the data harmonization level at their stations but, for example, they have never tried to perform a joint harmonization of the respective products.

Nevertheless, the authors smoothed the tone of the mentioned paragraph as follows: "*Federated networks set-up by international stakeholders (e.g. GALION – GAW Lidar Observation Network) are slowly evolving towards the harmonization of the different practices adopted within each of the federated networks (e.g. EARLINET, MPLNET, ADNET, LALINET), and, therefore, towards the homogeneity of the respective measurements and products; at present only one example of a coordinated monitoring of a global scale event (Nabro volcanic eruption) has been provided in literature (Sawamura et al., 2011).*
*It is useful for the scientific community to understand to what extent automatic lidars and ceilometers (ALCs) are able to provide an estimation of the aerosol geometric and optical properties and fill in the geographical gaps of the existing advanced lidar networks, like EARLINET,…..*".

The second paragraph is indeed saying that using commercial lidars/ceilometers, some effort must be spent to learn how these systems can fill in the observation gaps, also in terms of providing a support to the global lidar data harmonization.

58. "have been already investigated". That's fine, but in the next sentence or soon thereafter, you need to explain what the new contribution of this paper is.

At lines 88-89, the authors added the following paragraph: "Given the role commercial lidars and ceilometers may cover as a low-cost and low-maintenance baseline component of the aerosol non-satellite observing system at the global scale, several intercomparison experiments must be designed to assess the performances of commercial systems with respect to advanced multi-wavelength lidars and to ensure comparability between different instruments, measurements and retrieval techniques. Recommendation outcome from these experiments can also strongly support the design of current and future networks for the aerosol observation and the monitoring of pollution. Behind this motivation, the INTERACT campaign was arranged and took place at CIAO……".

62. "retrieval . . . can be performed using the molecular backscattering profile". Please be more specific. You mean the calibration can be performed using the molecular backscattering profile in a region where there is negligible aerosol. Without these additions, the phrase "the retrieval can be performed using the molecular backscatter profile" sounds like the Raman or HSRL retrieval.

**The text has been modified accordingly.**

215. Each lidar instrument has its own version of the quantity being considered, all with different names: attenuated backscatter, range-corrected signal (RCS), and normalized relative backscatter (NRB). It's a little confusing, but with some effort I see why you made these choices. It would be helpful to have a paragraph (earlier than this) where all three quantities are described in one place and the reasons for using different quantities for each instrument are provided.

**Given that the difference between the RCS and the NRB is in a constant term, and given that a normalization is operated to compare the MiniMPL and CIAO lidars, the authors will make use only of the values of the RCS and of the attenuated backscatter, which have been defined in the text of the new manuscript version.**

275. Please explain the normalization further. Is the MiniMPL normalized to match PEARL in the normalization region on a profile-by-profile basis for every profile?

**Each single MiniMPL profile is normalized to match CIAO lidars in the normalization region on a profile-by-profile basis for every profile. This has been further clarified in the text of the new manuscript version.**

322. If you mention after-pulse correction as a possibility, I think it needs to be supported. Otherwise it just sounds random and speculative.

**Indeed, this is a speculative discussion given that the the MiniMPL data processing has been performed by the manufacturer. Tests to assess the effect of a wrong after pulses correction have been performed by the authors though we agreed that the manufacturer shall investigate this hypothesis.**

325. Are the 12 cases all the measurements available from the whole six months deployment period, or have these been selected from a larger dataset? (Were the lidars operating continuously?)

**The MiniMPL, the CL51 and the CS135 were operated on a continuous basis from the respective deployment dates at CIAO, except for a few weeks when the MiniMPL had thermalizing problems and for a few days when the CS135 had communication issues. The CIAO EARLINET lidars, PEARL and MUSA, have been operated according to the EARLINET measurements schedule (3 night time measurements per week only with clear sky). This is the main reason why the number of cases is restricted though the automatic lidars and ceilometers were operated on a continuous basis.**

325. RCS or normalized relative backscatter? I thought that RCS meant non- normalized signals, so they could not be compared between two instruments? If there's no useful distinction in the terminology, it would be better to just use one name instead of three.

**To the authors knowledge, the term normalized-relative-backscatter (NRB) signal is in use within MPLnet (Micro-Pulse Nidar NETwork) which is defined by the equation:**

$$P_{\mathrm{NRB}}(r) = C[\beta_M(r) + \beta_P(r)]T_M^2(r)T_P^2(r).$$

**which indeed is equivalent to the RCS. According to the information provided by the manufacturer, it appears to the authors that there was a difference between RCS and NRB in the definition of the constant "C" of the lidar equation. However, as already mentioned above, to avoid misunderstandings, given that MiniMPL vertical profiles were normalized with respect to CIAO lidars profiles, in the new version of the manuscript the authors make use only of the values of RCS.**

338. "The good stability of the MiniMPL calibration ... is shown by the small variability (10%) of differences in the normalization region". I have a major problem with this statement. This one must be addressed. Aren't the profiles for all 12 cases normalized to the Raman lidar in the normalization region? So, for each profile, the normalization constant is divided out and each profile is independently set to have zero average difference in the normalization region. That means that to assess the profile-to-profile variability in the normalization constant, you'd have to explicitly look at the 12 normalization constants. That information is not present in the data shown in Figure 5. Variability in the normalization region in Fig 5 is only representative of high-frequency noise within the normalization region, so it informs you about the precision, but not about the stability over time.

**The authors agree with the reviewer. The numbers reported in the new version of the manuscript have been calculated using the real variability of the values for the normalization constant. The authors also remarked that these values have been calculated on small datasets, due to the number of available cases. The text of the new version of the manuscript has been refined accordingly.**

Figure 7. The differences in the scatter plots are very hard to make out given the data being compared are in two different plots. A figure showing both in the same figure would be better. For example, consider making a scatter plot of CL135 vs. MUSA/PEARL attenuated backscatter directly, and color code by extinction or stratify different ranges of extinction into multiple sub-plots. Accompanying them with another set color coded or stratified by altitude would also be helpful, I think, given that the interpretations of this figure in the text are related to specific altitude regions (the overlap region and the free troposphere). (Same comment for Figure 13.)

**Figure 7 and 13 have been modified according to the reviewer's suggestions.**

348. You say that the choice of aerosol extinction for the y-axis of Figure 7 (and 13) is to reveal differences in sensitivity to different aerosol types. In fact, you have not mentioned different aerosol types in your interpretation at all. Indeed, this task would be quite difficult with the information given in the figures since the relationship between attenuated backscatter and aerosol extinction is related not just to lidar ratio (indicator of aerosol type) but also to the amount of attenuation, and the attenuation may be a more dominant effect in this data set. If you really wanted to distinguish different aerosol types, you might consider including lidar ratio (from the Raman lidar) in the analysis. If you don't care about aerosol type, then probably just delete the statement about them.

**The term "aerosol types" has been deleted.**

354. "The most evident differences between the two lidars can be identified for values of extinction larger than about 5.0x10−5m−1 where miniMPL shows a broader scatter." It's very difficult to see this. I see that miniMPL has a bit more data at the low end of the x-axis and MUSA has a bit more data at the high end of the x-axis, but it is by no means obvious.

**The two panels in Figure 7 have been replaced using a plot simultaneously showing CIAO lidars and MiniMPL using different colors. This clarifies where are the differences between the RCS values measured by the two lidars.**

355. "Described above" What does this refer to, the unexplained statement about after- pulse correction?

**Yes, now this is clarified in the text.**

371. SNR. There seems to be some confusion between signal level and signal-to- noise ratio. The text says the CS135 SNR decreases above 3500 and the CS 51 SNR is higher. To me, it looks like the signal of CS135 decreases and the signal of CS51 is higher, but the noise in the CS51 signal is also quite high and it clearly does not agree with the more reliable Raman lidar, so it's likely this higher signal is an artifact. Your graph doesn't show SNR explicitly enough to aid in analyzing the SNR. I think you would have to look at both signal and noise and analyze noise levels explicitly to be able to make quantitative statements about how the SNR for the two instruments compare. From figure 8 I think you can say "The CS135 signal strongly decreases . . . .. The CL51 signal is higher but the noise suggests that it is not reliable to detect the residual aerosol. . ."

**The paragraph has been modified accordingly.**

Figure 10. Please check the units. Is the exponent -6? Or -5? Compare to Figure 7 which I think should be the same PEARL profile.

**The exponent is -6. This mistake has been fixed in the new version of the manuscript.**

422. Similar to my comment at line 338, I don't agree with this. Is each case normalized separately? If so, then the variability of the normalization constant is not represented in this plot. The error bars are related to the amount of variability over the few hundred meters of normalization range, but not to the stability of the normalization constant over time.

**Please see previous comments for CIAO lidars and MiniMPL.**

425. "The standard deviation of the normalization constant." Is this calculated separately by keeping track of the individual profile normalization constants? That's the correct way to do it.

**Please see previous comments for CIAO lidars and MiniMPL.**

439. "better performance of the CL51 when the values of extinction are larger for corresponding small values of backscatter and therefore indicates its improved SNR in the FT". Is it really true that these large values of extinction with corresponding small values of backscatter are in the free

troposphere? Wouldn't small backscatter values in the free troposphere more likely be accompanied by small values of extinction? It seems more logical that if there are small backscatter values when the extinction is large, that means there is significant attenuation, so the points are more likely low in the atmosphere below significant aerosol layers. This is important to check and clarify since you seem to be drawing a major conclusion (better performance of CL51 in the FT) almost wholly from this subtle and hard-to-interpret pattern.

**The manuscript refers only to night time data when the boundary layer height is very low and it is very often difficult to identify it with the MUSA-PEARL data. Therefore, most of the data and the reported value effectively corresponds to value in the FT. This is also consistent with the CIAO boundary layer climatology and modelling estimation of the nocturnal boundary layer. However, to avoid misinterpretations, the authors replaced FT with the term "aerosol residual layer" which improves the reader understanding.**
**The paragraph is now as follows: "**_these threshold values reveal the slightly better performance of the CL51 when the values of α are larger for corresponding small values of β' and, therefore, indicates CL51 improved SNR in the night time aerosol residual layer, in particular below 2.0 km asl where the profiles measured by both the ceilometer may be still affected by the correction for the incomplete overlap_**"**

443. "The overall stability of ceilometers' calibration constant . . .has been addressed in a statistical sense." I don't see any analysis of the overall stability of the calibration constant, see comments at line 338 and 422.

**Please see previous comments for CIAO lidars and MiniMPL.**

464. "in general is embedded" – please be more specific. Do you mean "is directly proportional to"? If so, please say that. If the relationship is more complicated than that, please include the equation.

**The sentence has been modified as follows: "**_The number of lasers pulses is included as a multiplying factor in the CHM15k data processing and it is one of the factors contributing to the so-called lidar constant (i.e the constant depending only on the lidar system experimental setup) within the lidar equation._**".**

470. "the calculated embedded constant". Does this mean lidar constant? Please say that. I think it would be good to put in some clarification that the calibration constant is an operational assessment of the lidar constant (which may have some noise or error). So, I think what you're saying is that if the true lidar constant has seasonal variability but a calibration constant is only assessed infrequently, then there will be a systematic error in the calibration constant.

**The authors thank the reviewer for noting this inconsistency. The text of the manuscript has been modified as follows: "**_This indicates that, across a fixed calibration range (i.e an aerosol free range to perform the molecular calibration), the normalization constant will range with a behaviour similar to that shown by the laser pulses in order to correct for the change in transmitted energy. As a consequence, given that the normalization constant is an operational assessment of the lidar constant plus a residual uncertainty due to the noise, also the true lidar constant will have seasonal variability. The reported ......._**".**

471. "what was reported during INTERACT". What was reported? Be more specific.

**With regard with the last two comments above, at line 470-471 of the old manuscript version, the authors have modified the paragraph as follows:** "*This indicates that, across a fixed calibration range (i.e. an aerosol free range to perform the molecular calibration), the normalization constant will range a behaviour similar to that shown by the laser pulses in order to correct for the change in transmitted energy. As a consequence, given that the normalization constant is an operational assessment of the lidar constant plus a residual uncertainty due to the noise, the true lidar constant will have the same seasonal variability as the normalization constant. The reported laser pulses variability can contribute to explain the large variability of the calibration constant (about 58 %) calculated during the six-month period of INTERACT-I (Madonna et al., 2015) which was only partly due to the variability of MUSA reference lidar (19%). During INTERACT-I, a direct correlation between the variability of the calibration constant and the seasonal temperature changes was found to be limited ($R^2$=0.6). Nevertheless, the seasonal change in the absolute value of the calibration constant was quite evident and addressed to the coupling of two simultaneous effects (temperature change and decrease in the aerosol loading). The reported seasonal variability of laser pulses also confirms that a calibration constant assessed infrequently will increase the systematic uncertainty contribution. It is possible to estimate over a period longer than 6 months a systematic uncertainty in the calibration constant of 10-20 %; over a period of three months the additional uncertainty may reduce to 5-10%.".*

471. "This partly explains". To me this finding of a temperature dependence suggests a hypothesis, but I don't see any testing or exploration of the hypothesis. Is there any indication that the variability during INTERACT was correlated with temperature? (I see in the earlier paper it was believed that there was, but there was no quantification of the correlation, and that information is missing entirely from this paper).

**To clarify this point, the authors has modified the corresponding paragraph as reported in the reply to the previous comment.**

471. As your continuing discussion points out, it doesn't seem that the size of the effect matches well at all. If the lidar constant is linearly related to the number of laser pulses, then the variabilities are also linearly related, and so 10% variability in pulse count can hardly explain 58% variability in the calibration constant. I think maybe it would be best to change the wording to remove or further deemphasize the "This partly explains" clause. While I agree that you have demonstrated that operators must be aware of temperature as a source of variability, as an investigation of the cause of the observed variability in the INTERACT observations, this is inconclusive at best.

**Please see previous comment.**

486. "most of the difference could be reduced after a reevaluation of the overlap correction". This statement in the conclusions is quite a bit stronger than the statement in the body of the text. In the text you demonstrated that reduction of the error was possible for a single case when the Raman lidar is available to show the true shape of the overlap region, but that it couldn't be corrected in most cases.

**To properly evaluate the overlap correction, an observation scenario with a low aerosol content is required which was not very common during the period of INTERACT-II. Nevertheless, the results of this test, along with the experience gained in the overall data analysis, allowed us to be optimistic on the possibility to improve the MiniMPL performances if a more robust evaluation of the overlap corrections function is carried out.**

**The sentence commented by the reviewer has been modified as follows: "*The RCS values measured with MiniMPL and CIAO lidars agree within 10-15 % and a re-evaluation of the overlap correction applied in the data processing could further reduced the discrepancies.*"**

492. "The CL51 is able to detect the molecular signal in the free troposphere". I'm not convinced this was demonstrated.

**Also on the basis of the results reported in the manuscript, the sentence has been smoothed as follows: "T*he CL51 appears to have the capability to detect the molecular signal in the free troposphere and, therefore, in order to retrieve the aerosol backscattering coefficient, the calibration of the attenuated backscatter using a molecular profile as a reference can be attempted over integration times longer than 1-2 hours.*"**

500. Since the introduction suggested a main motivation was "to understand to what extent automatic lidars and ceilometers are able to provide an estimation of the aerosol geometric and optical properties and fill in the geographical gaps of the existing advanced lidar network", it would be good to see some conclusion about this question here. You have said earlier "the only possible CL51 normalization to provide a reliable estimate of attenuated backscatter profile must be performed over a profile of attenuated backscatter from a reference lidar (like MUSA or PEARL)." These seems to argue against the usefulness of ceilometers for filling in existing gaps. Whether or not I am correctly guessing your conclusion, some discussion belongs in the conclusion section.

**The following paragraph has been added to the conclusions: "*The experience gained during INTERACT-I and INTERACT-II confirms the ceilometers' good performances in the qualitatively monitoring of aerosols in the boundary layer with enhanced profiling capabilities in the free troposphere only for the most advanced models. Nevertheless, the retrieval of aerosol attenuated backscatter (and of any related optical properties) appears to be affected by the instrumental issues which must be improved by the manufacturers in cooperation with the scientific community. It is possible therefore to argue that, compared to automatic (backscatter) lidars, more expensive but more powerful, the capability of ceilometers of filling in the existing observational gaps in lidar networks is continuous improving but it is still limited.*".**

3 Technical & grammatical:
143. Is it 16 optical channels? The description in the following sentences seems to say 16, not 17. Is something left out or is there a typo, maybe?
231. Probably "temperature" rather than "thermostat". A thermostat regulates temperature.
235. Instead of using "beta", spell out attenuated backscatter or use the symbol $\beta'$ that was already introduced.
344, 353, 354, elsewhere? Fix formatting of numbers in scientific notation
367. Possible missing word "between" 2.5 and 3.5 km asl
416. Delete the word "average"? I think you probably are reporting the standard deviations of the fractional differences, not the standard error of the mean. If you are reporting the standard error

of the mean, please use that terminology rather than "standard deviation of the average".

448. Replace indifferently with interchangeably

451. "over the time", delete "the"

504. "INTERACT-II". Should this be "INTERACT-I"?

Figure 1. A log scale might be more informative for this quantity.

Figures 3, 4, 6, 8 . The label "LIDAR" should be "MUSA", "PEARL" or "MUSA/PEARL"

Figure 7. the axis labels are really small and it's not possible to zoom them in enough to make them clear. It would be good to remake these with bigger axis labels. (But see above: I also have a suggestion for a different plot style altogether.)

Figure 7 caption. Please state the time & date of the comparisons.

Figure 8. "Using t[w]o normalization ranges (below 3 km and above 8 km)". It appears that this is incorrectly pasted from another figure. Figure 8 doesn't seem to have two normalization regions.

Figure 11 caption, line 841. "standard deviations of the fractional differences" not "average", I think (see above)

**All the technical corrections have been fixed in the new version of the manuscript according to the reviewers' suggestions.**

---

## Author Comment (AC6) · 23 Jan 2018

Reply to the comments by the anonymous Referee #1

**The authors gratefully acknowledge the reviewer's effort in improving the quality of the manuscript. Below, a point-to-point response is provided to all of the reviewers' major comments. The remaining minor comments have been all of them fixed in the new version of the manuscript.**

General Comments

The paper "Intercomparison of aerosol measurements performed with multi- wavelength Raman lidars, automatic lidars and ceilometers in the frame of INTERACT- II campaign" reports the results of a campaign using a variety of instruments to measure aerosol in cloud-free or clear-sky conditions. While the authors report interesting results, I think that they could make the analysis more rigorous and motivate the work more clearly. I have made recommendations below.

Specific Comments

1. I recommend that the authors provide general motivation in the introduction for this study. Why does anyone need to measure atmospheric aerosols using these types of instruments? Why is this intercomparison needed? Is it to help design better networks for measuring pollution, for example? I would like to understand this and to make sure the audience understands how the intercomparison gives us important and useful information. Can the authors say anything specific about the aerosols that were measured (type or other properties) during the campaign?

**Though the introduction largely describes the current state-of-the-art for the use of ceilometers for aerosol profiling in the troposphere and their big potential to improve the current baseline aerosol observing capabilities at the global scale as low-cost and low-maintenance instruments, to meet the reviewer' s request, in the new version of the manuscript, the two following sections have been added to the introduction at line 69: "_Given the role commercial lidars and ceilometers may cover as a low-cost and low-maintenance baseline component of the aerosol non-satellite observing system at the global scale, several intercomparison experiments must be designed to assess the performances of commercial systems with respect to advanced multi-wavelength lidars and to ensure comparability between different instruments, measurements and retrieval techniques. Recommendation outcome from these experiments can also strongly support the design of current and future aerosol observing networks for measuring aerosols and pollution._".**

2. Please can the authors explain, again in a general way, which of the instruments is expected to measure aerosols (of a given type) most accurately and why. For example, can you give a general sense of where (in the atmospheric column) the instruments are expected to give the best results? And why? Perhaps it would be helpful to touch on differences in wavelength here as well as other differences in hardware or firmware? I realise that none of the instruments gives us "truth", but can the authors give the reader a sense of the accuracy expected? Thus, when the differences are reported, the readers immediately understand which of the instruments is believed to be closer to the true observed quantity.

I suggest these two points in order to give the reader a better sense of why these particular instruments are important to study (as I think that they are) and to make a stronger case for why the intercomparison analysis in this paper matters to the community.

**The following paragraph has been added at the lines 58-70 of the new version of the manuscript to meet the reviewer's request of clarification for the reader.:**

*"With respect to the past when lidars were strictly research instruments, many modern automated lidars are available on the commercial market and can now contribute efficiently to continuous monitoring atmospheric aerosol. Automatic lidars have very different features from models equipped with diode-pumped laser or solid-state laser emitting in the UV at 355 nm or in the visible spectrum at 532 nm. Only multi-wavelenght lidars emits wavelengths in the near infrared at 1064 nm. Typically, the higher is the energy emitted per laser pulse (In the order of a few μJ to mJ) the more demanding will be be the required maintenance and costs. In analogy, higher is the energy emitted per laser pulse the better will be the lidar signal to noise ratio and the lower will be the random uncertainty affecting the estimation of the estimated aerosol properties. A ceilometer generally differentiates from a one-wavelength automatic lidar because it emits a single wavelength in the near infrared between 900 and 1100 nm to avoid strong Rayleigh scattering, the pulse repetition rate is on the order of a few kilohertz, and the pulse energy of the laser is in the order of a few μJ to allow eye-safe operations, continuously and unattendedly operations. UV and visible automatic lidars can typically cover the whole tropospheric range while ceilometer, depending on the model, may cover the boundary layer only or detect aerosol features also in the free troposphere."*

**We want to clarify that the author intentionally didn't provide any details on the system precision and accuracy because these may strongly change from an instrument to another and they prefer to provide an extensive characterization of the measurements in the section where the intercomparison with the CIAO lidars is discussed.**

3. In section 5 MUSA is referred to as the reference signal in the full overlap region. Why is MUSA the reference? Is it expected to be the highest standard of measurement to which we want to compute the ceilometer observations?

**In the new version of the manuscript the following paragraph has been added to explain why MUSA is considered the "reference" system for the intercomparison campaign:** *MUSA is routinely tested with respect to several systematic quality-assurance tests developed in order to harmonize the lidar measurements, to set up quality standards, and to improve the lidar data evaluation (Pappalardo et al., 2014). MUSA signals are also routinely evaluated using the Rayleigh fit test, and signal-to-noise analysis described in Baars et al. (2016). Additionally, the telecover test (Freudenthaler, 2008) is performed regularly and especially after transportation of the system. The system is aligned using a CCD camera to reduce the effect of misalignment between the telescope and laser axis, being MUSA a bistatic lidar. Finally, the multi-wavelength detection capability enables to so called "3+2" lidar data analysis which, taking advantage of the simultaneous retrieval of lidar extensive (aerosol extinction at 355 nm and 532 nm; backscattering coefficients at 355 nm, 532 nm and 1064 nm) and intensive properties (lidar ratios at 355 nm and 532 nm and color ratios) at different wavelengths permits to check the physical consistency of the retrieved aerosol properties.*

**The authors also clarified when describing PEARL lidar that "PEARL has been extensively intercompared with MUSA to have a redundant aerosol profiling capability at CIAO.".**

4. Please define the "fractional difference". For example, in section 4 Paragraph 5, "average fractional difference" is not defined and later in the paragraph (line 327) an "average difference" is increasing. Are these the same metric? The authors need to define clearly the measure or measures of difference applied to the results.

**The concept of fractional difference is now explained at lines 268 and 269 using the following sentence: "_Fractional difference is defined as the difference between CIAO lidar and MiniMPL RCS values normalized to CIAO lidar RCS_".**

5. There are a few places where the authors discuss "random uncertainty" (section 4 for example in line 322). Please could the authors define how they determine the random uncertainty? Also, if there are some statistical tests being performed to assess differences then please state which tests are being used. For example, is there a null hypothesis of random white noise?

**In the new version of the manuscript a reference has been added at the corresponding lines to clarify the processing applied to the CIAO lidar signals and the retrieval of the corresponding uncertainties.**
**Random uncertainty is the contribution to the total uncertainty budget typically named by lidar experts as "statistical error." According to the GUM and metrology, the term "error" is less appropriate than uncertainty when an estimation of the error is provided. According the GUM (Guide to the Expression of Uncertainty in Measurement):**
**"_Whereas the exact values of the contributions to the error of a result of a measurement are unknown and unknowable, the uncertainties associated with the random and systematic effects that give rise to the error can be evaluated. But, even if the evaluated uncertainties are small, there is still no guarantee that the error in the measurement result is small; for in the determination of a correction or in the assessment of incomplete knowledge, a systematic effect may have been overlooked because it is unrecognized. Thus, the uncertainty of a result of a measurement is not necessarily an indication of the likelihood that the measurement result is near the value of the measurand; it is simply an estimate of the likelihood of nearness to the best value that is consistent with presently available knowledge._**
**_Uncertainty of measurement is thus an expression of the fact that, for a given measurand and a given result of measurement of it, there is not one value but an infinite number of values dispersed about the result that are consistent with all of the observations and data and one's knowledge of the physical world, and that with varying degrees of credibility can be attributed to the measurand._".**
**The random uncertainty for raw lidar signals is evaluated as the standard deviation of the Poisson distribution of counts (square root of the counts), because the backscattered radiation is acquired in photon-counting mode and a Poisson distribution is assumed for the detected signals. For CIAO lidars, the raw signals are pre-processed to apply instrumental corrections and, optionally, a vertical smoothing or temporal averaging. This stage is commonly known as "pre-processing" of raw signals. The pre-processed signals, with time and vertical resolutions depending, respectively, on temporal and vertical integration performed by the pre-processing module, are the input of the second part of the processing algorithm, known as "processing" of the pre-processed signals, providing the profiles of aerosol optical properties. These profiles have a time sampling which is the integration time used in pre-processing stage and effective vertical resolution depending on the vertical smoothing performed in pre-processing and processing modules.**

**The random or statistical uncertainties of pre-processed signals are calculated starting from random uncertainties of raw lidar signals, by using the standard formula of statistical uncertainty propagation at each step of the pre-processing stage. Random uncertainties in the aerosol extinction or backscattering profiles are calculated starting from random uncertainties of pre-processed lidar signals, by using the Monte Carlo simulation for all applied signal handling procedures in the processing stage.**

**For the MiniMPL, though this is a polarized elastic backscatter lidar operating only at 532 nm, the applied processing follows a similar logic in the pre-processing of the lidar signals.**

6. At the end of the technical corrections, I have placed a number of comments on the figures which need to be addressed.

**The authors fully addressed all of the technical corrections recommended by the Reviewer #2 and provide below comments to the most relevant.**

Technical Corrections
Title: Please change "frame" to "framework". Text:
1. Many acronyms are undefined in the main body of the paper. To aid the reader, please explicitly define the following: CNR-IMAA, EARLINET, FOV, FWHM, GRUAN, RAOB, HYSPLIT and APD in line 149 (is it Avalanche Photo Diode?) used before line 161 Avalanche Photo Detector are these the same "APD"?,

**The listed acronyms have been defined in the new version of the manuscript.**

2. Please put units on the RCS. I believe that the authors are using "arbitrary units" (a.u.) throughout. Is this correct? Can a.u. be placed next to all the measurements please?

**Ok. A.u. has been reported next to all the measurements only when absolute values of RCS are reported.**

Line 22 Is average difference a root mean squared difference? Absolute difference? Or something else?

**This is "average fractional difference" and has been appropriately modified throughout the manuscript.**

Line 29 Rewrite to something more like: "Some tests performed during this campaign using the CHM15k ceilometer made it clear that the CHM15k historical dataset (2010- 2016) available at CIAO should be reviewed in order to evaluate the potential effect of…."

**Modified accordingly.**

L239 Please could you briefly (in a sentence) say why the assumption of < 1% is a good one? I can see there is a reference, but a quick explanation would be helpful, if a brief one is possible.

**Upon the basis of additional calculations, the authors have modified the text as follows "The uncertainty contribution for the spectral dependence of β' and, therefore, of the aerosol backscattering coefficient and of molecular and aerosol extinction coefficients has been estimated within a few percent.".**

L 352 Is the output profile from Raman PEARL lidar? If so, is it interpolated to the same resolution as the RCS from which instrument?

**Figure 7 embeds the measurements performed with both MUSA and PEARL, all of them interpolated at the same output resolution.**

L377-378 "..because MUSA is considered the reference signal only in the full overlap...." Has this been stated before? Has MUSA been the reference all along?

**Please see the authors reply to the reviewer's general comments.**

L394 I would suggest the wording should be changed to "Dark current measurements or profiles", not just "dark currents"

**The authors have discussed the use terminology and it seems that the community working with ceilometers prefers this terminology. This is the reason why kept it also because we believe this is not confusing.**

Figures:
1. All sub-panels within all figures should be labelled with letters a,b,c, etc. 2. In the text and captions all of the sub-panels in the figures should be referred to using the figure number and letter together. Please do not use left/right, top/bottom. The letters make the text concise and precise. For example, caption for figure 8 should read more like: "Panel a shows attenuated backscatter retrieved from ... Similarly, panel b shows the same comparison but for 01 December ..."
Figure 3, 4,6, 8, 9, 12 have a red line (or red bar) labelled "Lidar" but MUSA is in the caption. Lidar is not specific enough. Please make the legend consistent and more precise. Is it MUSA Lidar? In contrast, for example, Figure 10 has a red line called PEARL which is also a Raman lidar like MUSA.
Figure 4. Caption is confusing. Can authors please explain what they mean by "using NOAA HYSPLIT model started at the three levels from the ground the top layer ob- served by MUSA and MiniMPL lidars"? Are we talking about model levels? What is "the top layer observed"?
Figure 5. Should read "Blue line is the same as the black line but..." Also, captions usually put the line colour or line style in parentheses like this: "Profiles of the average fractional difference (black line)..."
Fig 9 End of caption: " Panel b shows the attenuated backscatter vertical profiles taken using the MUSA/PEARL lidar which operates at wavelength 1064 nm during the same time period as was used to create the average profiles in panel a."
Fig 10 Change to "Comparison between" not among. Also the line colour is "green" not "dark". This is the line with the dark current measurement subtracted away but the line is green.
Fig 11 Change "calculated on" to "calculated for"
Figure 14: What time does each square represent? Can't be 30 s resolution?! There are 7 years on the x-axis. How were the laser pulses averaged?

**All the Figures have been modified according to the reviewers' suggestions. For the last question about Figure 14 each square represents the number of pulses emitted per hour.**

---

## Author Response (AR2)

**Response to the Reviewers of the paper "Intercomparison of aerosol measurements performed with multiwavelength Raman lidars, automatic lidars and ceilometers in the framework of INTERACT-II campaign" by Madonna et al.**

**Reviewer#2**

The paper "Intercomparison of aerosol measurements performed with multiwavelength Raman lidars, automatic lidars and ceilometers in the framework of INTERACT-II campaign" has been greatly improved by revision. The reviewer particularly appreciates revisions to the figures and the analysis of the stability of the normalizations. My remaining suggestions are small and don't reflect major problems with the quality of the work. I would like to repeat that the organization and flow of this paper are particularly clear and concise. On the other hand, I would like to make sure that it undergoes a thorough edit for English language and grammar, because there are a number of missing words and awkward phrases at the sentence level (mostly not listed here); however, these mostly don't interfere with understanding the authors' meaning.

**The authors thank both the reviewers for his/her useful cooperation during the review process aiming at the improvement of the quality of their manuscript. The authors also carried out an additional and more detailed English review by one of the mother language co-authors. Comments by the authors are reported below in bold.**

Some specifics:

Lines 487-496, the evidence in Figure 13 supporting the conclusion about CL51 having better SNR than CL135 still seems rather weak.

**The sentence has been rephrased as follows: "*For the CL51, differences with CIAO LIDARs in the scatter plot are small and mainly related to the region where $\beta' < 5.0\ 10^{-7}\ m^{-1}\ sr^{-1}$ and $\alpha par > 8.0\ 10^{-5}\ m^{-1}$: in this region, the values observed by CIAO LIDARs correspond to very small values detected by the CL51. For the CS135, though a small number of cases are available, a behavior similar to the CL51 can be identified in the region where $\beta' < 6.0\ 10^{-7}\ m^{-1}\ sr^{-1}$ and $\alpha par > 5.0\ 10^{-5}\ m^{-1}$; these threshold values reveal the slightly better performance of the CL51 when the values of $\alpha par$ are larger for corresponding small values of $\beta'$. These values are measured within the night time aerosol residual layer, in particular below 2.0 km asl where the profiles measured by both the ceilometers may be still affected by the correction for the system incomplete overlap.*"**

"lower aerosol loading". Since the data is attenuated backscatter with an incomplete overlap, is it really possible to say that the aerosol loading is lower? Maybe it would be better to say lower signals or lower RCS.

**The text of the manuscript has been changed using "lower RCS values" instead of "lower aerosol loading".**

Misuse of "respectively" such as at line 17 and 387. "Respectively" should link two lists of items. It has no meaning if there is only one list. Examples of correct usage occur at lines 293 and 595.

Multiple grammatical errors at lines 38-39 probably from cut-and-paste.

At line 74, consider breaking the sentence, replacing "profiles and relies" with "profiles. The calibration relies". I think this makes it clearer that the later part of the sentence ("can be performed") refers to the calibration rather than the retrieval as a whole.

112, "The campaign" should be "The INTERACT" campaign (I think).

and 321, replace "particle backscattering" with "particle attenuated backscattering" and replace "total backscatter radiation" with "total attenuated backscatter radiation".

"maximum value" is really "maximum average value"

**All the minor comments reported above have been changed according to the reviewer's suggestions.**

322-324, Please clarify, are the data shown in Figure 1 already normalized?

**The time series reported in Figure 1 are representative of the RCS calculated from the raw signals for both the lidars, MUSA and MiniMPL. Therefore, the two plots are not normalized and provided only for a initial qualitative comparison which is then addresses on quantitative basis in the data analysis discussed in the manuscript. This has been further clarified in the text.**
* * *
**Reviewer#1**
The article (amt-2017-399:Intercomparison of aerosol measurements performed with multi-wavelength Raman lidars, automatic lidars and ceilometers in the frame of INTERACT-II campaign) is very good scientifically and of interest to the community.

I have only one concern. There are problems with incorrect grammar that should be addressed before publication. I list a few examples below, but one of the authors needs to do a thorough check of the whole paper and re-write where necessary.

For example, the first two sentences of the introduction need a re-write.
Suggestions:

1. The monitoring of Essential Climate Variables (ECVs) using low-cost and low-maintenance automatic systems represents a challenge for the scientific community and instrument manufacturers over the next decade.

2. Automatic lidars do not report progress. They can report a signal. Perhaps authors mean something like: The use of automatic lidars for the vertical profiling of aerosol properties both in the boundary layer and in the free troposphere has progressed steadily over the last few years.

**Both the sentences above have been modified according to the reviewer's suggestion. In addition, as mentioned above, the authors carried out a further and more detailed English review of the manuscript.**

There are also errors in the paper:

L18 "The campaign HELD"
L 60 "monitoring OF atmospheric aerosol"
L62 "Typically, THE higher..., THE higher…"

**All the other minor errors have been fixed in the new version of the manuscript.**

[revised manuscript text omitted]